# FlowBypass: General Training-Free Rectified Flow Image Editing via Trajectory Bypass

## Abstract

Training-free image editing has attracted increasing attention for its efficiency and independence from training data. However, existing approaches predominantly rely on inversion–reconstruction trajectories, which impose an inherent trade-off: longer trajectories accumulate errors and compromise **fidelity**, while shorter ones fail to ensure sufficient **alignment** with the edit prompt. Previous attempts to address this issue typically employ backbone-specific feature manipulations, limiting general applicability. To address these challenges, we propose **FlowBypass**, a novel and analytical framework grounded in Rectified Flow that constructs a bypass directly connecting inversion and reconstruction trajectories, thereby mitigating error accumulation without relying on feature manipulations. We provide a formal derivation of two trajectories, from which we obtain an approximate bypass formulation and its numerical solution, enabling seamless trajectory transitions. Extensive experiments demonstrate that FlowBypass consistently outperforms state-of-the-art image editing methods, achieving stronger prompt alignment while preserving high-fidelity details in irrelevant regions.

## 1 Introduction

Image editing has become a powerful paradigm for controlling and manipulating visual content through intuitive instructions and prompts. Among various approaches, *training-free image editing* has gained increasing attention because it avoids the need for large-scale data or costly fine-tuning while still enabling flexible and effective manipulations. This property makes training-free methods especially appealing for practical deployment across diverse real-world applications.

However, a persistent challenge limits the effectiveness of existing training-free methods. Most existing approaches are built upon inversion–reconstruction trajectories, where an image is first inverted into noise and then reconstructed under the guidance of an edit prompt. This design inherently creates a trade-off: long trajectories accumulate estimation errors that erode **fidelity** in regions that should remain untouched, while short trajectories weaken **alignment** with the prompt, resulting in incomplete edits. Prior efforts to alleviate this trade-off include trajectory adjustments (Brack et al., 2024; Rout et al., 2025), prompt refinements (Mokady et al., 2023; Miyake et al., 2025), and feature manipulations (Hertz et al., 2023; Simsar et al., 2025). However, most of these methods traverse full trajectories, which exacerbates error accumulation near the inverted noise, and some rely on backbone-specific feature interventions, which hinder generality across generative models.

To overcome these limitations, we introduce **FlowBypass**, a general and analytical training-free image editing framework grounded in Rectified Flow (RF) (Liu et al., 2022). The central idea is a **trajectory bypass** that directly connects the inversion and reconstruction trajectories at intermediate states, thereby reducing accumulated estimation errors without relying on feature manipulations. This design not only addresses fidelity–alignment trade-offs but also ensures strong generalizability across diverse backbones. We develop FlowBypass starting from theoretical insights: we first analyze the inversion and reconstruction trajectories, then derive an approximate formulation of the bypass, and finally design a numerical solution via Euler discretization (Euler, 1768). This theoretical-to-practical pipeline enables efficient trajectory transitions while providing a unified solution applicable to a wide class of Rectified Flow models. Extensive experiments across diverse editing tasks demonstrate that FlowBypass consistently outperforms state-of-the-art training-free

methods, achieving stronger alignment with edit prompts while preserving high-fidelity details in irrelevant regions. Our main contributions are summarized as follows:

• **Theoretical Foundation of FlowBypass**: We provide a rigorous mathematical formulation of inversion and reconstruction trajectories, and derive an approximate bypass solution with a tractable analytical form, laying the theoretical groundwork for FlowBypass.

• **Unified and Practical Realization**: We transform the analytical solution into an efficient discretized form, yielding a unified training-free image editing framework based on Rectified Flow. It bridges theory and practice by bypassing inversion to directly guide reconstruction, removing the need for backbone-specific feature manipulations and improving both fidelity and alignment.

• **State-of-the-Art Performance**: Extensive experiments demonstrate that FlowBypass consistently achieves state-of-the-art results across challenging editing scenarios, delivering superior fidelity–alignment trade-offs and robust generalization compared to existing training-free methods.

## 2 RELATED WORKS

Training-free image editing aims to manipulate images without additional training or fine-tuning. Rather than updating model parameters (Brooks et al., 2023; Yu et al., 2025; Chen et al., 2025; Xiao et al., 2025), these methods directly exploit pre-trained generative models. Existing approaches can be broadly grouped into four categories according to how they modify the sampling process.

The most common paradigm is noise-inversion methods, the dominant paradigm where a origin image is mapped into the noise space and then reconstructed into the edited result. Most build upon DDIM-inversion (Song et al., 2021), with modifications to inversion and reconstruction trajectories (Huberman-Spiegelglas et al., 2024; Mokady et al., 2023; Brack et al., 2024; Miyake et al., 2025), but they remain fundamentally constrained by accumulated errors, especially near the inverted noise. Recently, training-free editing has also been extended to Rectified Flow (RF) models (Wang et al., 2025; Rout et al., 2025), but most remain rooted in DDIM-based principles, which continue to suffer from the accumulation of trajectory errors.

The second paradigm is prompt- or condition-refinement, which enhances the conditioning signal by incorporating information from the source image, thereby enabling edits within the reconstruction trajectory (Ravi et al., 2023; Wang et al., 2023). The third paradigm is feature manipulation, where intermediate representations are modified either by injecting features from the inversion trajectory to improve fidelity or by amplifying prompt-related activations to strengthen alignment (Hertz et al., 2023; Tumanyan et al., 2023; Cao et al., 2023; Simsar et al., 2025; Feng et al., 2025).

The fourth paradigm omits the explicit inversion process, instead relying on a coarse preliminary interception to perform a pseudo-inversion and subsequently operating directly in the image distribution, a substantially more complex domain than the noise-blended space (Xu et al., 2024; Kulikov et al., 2025). Consequently, inversion-free methods may introduce artifacts during editing, often resulting in degraded visual quality.

Despite their methodological diversity, most of these methods rely on full trajectory traversal, which inherently amplifies discretization errors and weak conditional guidance. As a result, training-free editing often introduces unintended modifications in irrelevant regions or even fails to realize the intended edit. This persistent bottleneck motivates the development of **FlowBypass**, a principled framework designed to mitigate accumulated errors while preserving both fidelity and alignment.

## 3 METHOD

### 3.1 PRELIMINARIES

**Rectified Flow (RF).** RF formulates image generation as a continuous transformation of a Gaussian noise distribution $\mathcal{N}(0, \boldsymbol{I})$ into the target data distribution $p_{\text{data}}$ under conditioning signals, governed by a velocity field expressed as an ordinary differential equation (ODE):

$$\frac{d}{dt}z_t = \bar{v}(z_t, t, C), \tag{1}$$

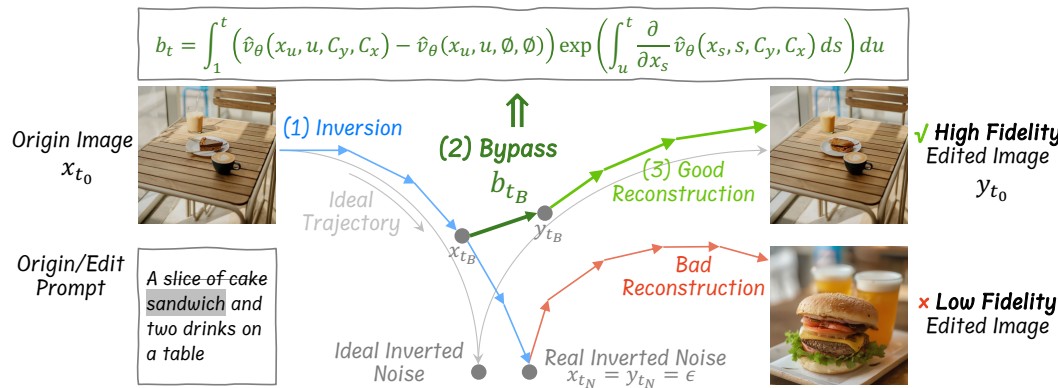

Figure 1: **Framework of FlowBypass.** FlowBypass consists of three steps: **(1)** Inverse the input image $x_{t_0}$ with Euler discretization to obtain the inversion trajectory; **(2)** Calculate the bypass $b_{t_B}$ using inversion trajectory according to Equ. 13; **(3)** Reconstruct from intermediate state $y_{t_B} = x_{t_B} + b_{t_B}$ to obtain edited image $y_{t_0}$. We show the differences between the origin prompt and the edit prompt by marking removed words with strikethrough and added words with a gray background.

where $t \in [0, 1]$ is the continuous timestep, $z_0 \sim p_{\text{data}}$ is a real image, $z_1 = \epsilon \sim \mathcal{N}(0, \boldsymbol{I})$ is a Gaussian noise sample, $\bar{v}$ is the marginal velocity field (Lipman et al., 2023), and $C$ is the provided condition (*e.g.*, text prompts). The RF network $v_\theta$ is trained to approximate the marginal velocity field $\bar{v}$ by minimizing $\mathbb{E}_{t,z_0,\epsilon}\|v_\theta(z_t, t, C) - (\epsilon - z_0)\|^2$, where $z_t = t \cdot \epsilon + (1-t) \cdot z_0$ is the linear interpolation of $z_0$ and $\epsilon$. Sampling seeks to recover a clean image $z_0$ from noise $\epsilon$ using Euler discretization of Equ. 1, once the velocity field is well estimated by $v_\theta$.

**Classifier-Free Guidance (CFG).** CFG is widely used to balance semantic alignment and visual fidelity during sampling. Given velocity fields predicted under a positive prompt $C_p$ and a negative prompt $C_n$, the guided velocity field can be formulated as:

$$\hat{v}_\theta(z_t, t, C_p, C_n) = v_\theta(z_t, t, C_n) + \omega \cdot \Big(v_\theta(z_t, t, C_p) - v_\theta(z_t, t, C_n)\Big), \tag{2}$$

where $\omega$ is the guidance scale. The larger values of $\omega$ amplify the contribution of positive prompt, typically leading to improved semantic alignment with provided condition but at the risk of sacrificing diversity and visual fidelity. Substituting $\hat{v}_\theta$ into Equ. 1 yields the CFG-guided trajectory:

$$z_t = z_r + \int_r^t \hat{v}_\theta(z_\tau, \tau, C_p, C_n)d\tau. \tag{3}$$

**Numerical Integration.** The above ODEs can be solved numerically through discretization. Starting from $z_{t_N} = \epsilon \sim \mathcal{N}(0, \boldsymbol{I})$ at $t_N = 1$ and using a monotonic sequence of timesteps $\{t_0 = 0, t_1, \ldots, t_N = 1\}$, Euler discretization updates the state as:

$$z_{t_{i-1}} = z_{t_i} + (t_{i-1} - t_i) \cdot \hat{v}_\theta(z_{t_i}, t_i, C_p, C_n). \tag{4}$$

Iteratively applying Equ. 4, the initial noise gradually moves along the approximated trajectory until it reaches the final clean image $z_{t_0}$.

**Training-free Image Editing.** Given a pre-trained generative model $\mathcal{M}$, an input image $x_{t_0}$, and an optional origin condition $C_x$, the objective is to synthesize an edited image $y_{t_0}$ that satisfies a user-specified edit condition $C_y$, while preserving the original content not related to the edit condition. The dominant pipeline consists of two stages: (i) *inversion*, mapping $x_{t_0}$ into a corresponding Gaussian noise $x_{t_N} \equiv \epsilon \equiv y_{t_N}$ under the guidance of inversion condition $C_{inv}$; and (ii) *reconstruction*, resampling from the noise $y_{t_N}$ to reconstruct the edited image $y_{t_0}$ under the guidance of reconstruction condition $C_{rec}$. The inversion trajectory and the reconstruction trajectory share the same noise point $x_{t_N} \equiv \epsilon \equiv y_{t_N}$, while differing only in their image points $x_0$ (origin image) and $y_0$ (edited image). While the design of $C_{inv}$ and $C_{rec}$ varies across methods, all such approaches depend on traversing the full inversion–reconstruction trajectory. This reliance makes them inherently vulnerable to numerical error accumulation and attenuated conditional guidance—limitations that motivate the bypass strategy developed in our framework.

## 3.2 MOTIVATION

A fundamental challenge in inversion–based training-free image editing is the progressive accumulation of errors during inversion and reconstruction. These errors in inversion primarily stem from two sources: (i) numerical discretization when solving the ODE in Equ. 4, and (ii) intrinsic mismatch between inversion and forward sampling. Concretely, the inversion equation $x_{t_i} = x_{t_{i-1}} + (t_i - t_{i-1}) \cdot \hat{v}_\theta(x_{t_i}, t_i, C_p, C_n)$ reformulated from Equ. 4 contains $x_{t_i}$ on both sides, thereby necessitating an approximation of $\hat{v}_\theta(x_{t_i}, t_i, C_p, C_n)$ with either $\hat{v}_\theta(x_{t_{i-1}}, t_i, C_p, C_n)$ or $\hat{v}_\theta(x_{t_{i-1}}, t_{i-1}, C_p, C_n)$. Such approximation introduces errors that propagate along the trajectory, ultimately yielding a degraded terminal state $x_{t_N}$. While the reconstruction trajectory is unaffected by intrinsic mismatch, it is still subject to errors caused by numerical discretization.

Our key insight is that fidelity loss in training-free editing is largely attributable to reconstructing from this corrupted terminal noise $x_{t_N}$. This observation motivates a departure from conventional designs: rather than relying on the error-prone endpoint, we propose to bypass it by selecting an intermediate state $x_{t_B}$, which preserves higher fidelity due to reduced error accumulation. We then construct the edited counterpart $y_{t_B}$ by introducing a bypass term $b_{t_B}$, as illustrated in Fig. 1. This bypass enables a direct transition from inversion to reconstruction at $t_B$, effectively mitigating accumulated errors while ensuring semantic alignment with the edit prompt through $b_{t_B}$.

## 3.3 FORMULATION OF FLOWBYPASS

Given a pre-trained generative RF network $v_\theta$, an origin image $x_0$ with its associated origin prompt $C_x$, and a target edit prompt $C_y$ specifying the desired output $y_0$, two ODEs can be established as:

$$x_t = x_1 + \int_1^t \hat{v}_\theta(x_\tau, \tau, C_{inv}^p, C_{inv}^n) d\tau,$$

$$y_t = y_1 + \int_1^t \hat{v}_\theta(y_\tau, \tau, C_{rec}^p, C_{rec}^n) d\tau, \tag{5}$$

where $C_{inv}^p$ and $C_{inv}^n$ are the positive and negative prompts that control the inversion trajectory, and $C_{rec}^p$ and $C_{rec}^n$ control the reconstruction trajectory. The specific choice of these prompts is detailed in Sec. 3.4. We define the bypass $b_t$ as the offset between the two trajectories:

$$b_t = y_t - x_t = \int_1^t \hat{v}_\theta(y_\tau, \tau, C_{rec}^p, C_{rec}^n) d\tau - \int_1^t \hat{v}_\theta(x_\tau, \tau, C_{inv}^p, C_{inv}^n) d\tau$$

$$= \int_1^t \left( \hat{v}_\theta(y_\tau, \tau, C_{rec}^p, C_{rec}^n) - \hat{v}_\theta(x_\tau, \tau, C_{inv}^p, C_{inv}^n) \right) d\tau. \tag{6}$$

Differentiating both sides with respect to $t$, which leads to:

$$\frac{d}{dt} b_t = \hat{v}_\theta(y_t, t, C_{rec}^p, C_{rec}^n) - \hat{v}_\theta(x_t, t, C_{inv}^p, C_{inv}^n)$$

$$= \hat{v}_\theta(x_t + b_t, t, C_{rec}^p, C_{rec}^n) - \hat{v}_\theta(x_t, t, C_{inv}^p, C_{inv}^n). \tag{7}$$

Since $b_t$ can be regarded as a small offset, so that the first term $\hat{v}_\theta(x_t + b_t, t, C_{rec}^p, C_{rec}^n)$ can be approximated with First-Order Taylor Expansion Formula (Taylor, 1715):

$$\hat{v}_\theta(x_t + b_t, t, C_{rec}^p, C_{rec}^n) \approx \hat{v}_\theta(x_t, t, C_{rec}^p, C_{rec}^n) + \frac{\partial}{\partial x_t} \hat{v}_\theta(x_t, t, C_{rec}^p, C_{rec}^n) \cdot b_t. \tag{8}$$

Substituting Equ. 8 into Equ. 7 yields an approximate linear ODE for $b_t$:

$$b_t \approx b_t^*, b_1^* = 0,$$

$$\frac{d}{dt} b_t^* = \left( \hat{v}_\theta(x_t, t, C_{rec}^p, C_{rec}^n) - \hat{v}_\theta(x_t, t, C_{inv}^p, C_{inv}^n) \right) + \frac{\partial}{\partial x_t} \hat{v}_\theta(x_t, t, C_{rec}^p, C_{rec}^n) \cdot b_t^*, \tag{9}$$

where $b_t^*$ is an approximated $b_t$. Equ. 9 is a first-order homogeneous linear differential equation, whose analytical solution is:

$$b_t^* = \int_1^t \left( \hat{v}_\theta(x_u, u, C_{rec}^p, C_{rec}^n) - \hat{v}_\theta(x_u, u, C_{inv}^p, C_{inv}^n) \right) \exp \left( \int_u^t \frac{\partial}{\partial x_s} \hat{v}_\theta(x_s, s, C_{rec}^p, C_{rec}^n) ds \right) du. \tag{10}$$

This closed-form characterization of $b_t^*$ provides a principled means of estimating the bypass term. Rather than depending on error-prone terminal inversion states, FlowBypass analytically derives an offset that directly links inversion and reconstruction at any intermediate time $t$. This formulation alleviates accumulated discretization errors, ensuring fidelity in irrelevant regions and alignment with edit prompts, while remaining agnostic to backbone architectures, thereby providing a general and powerful framework for training-free image editing.

### 3.4 IMPLEMENTATION OF FLOWBYPASS

Since Equ. 3 and Equ. 10 involve integral calculations, Euler discretization is employed in the inversion, bypass calculation, and reconstruction to obtain numerical solutions. Following Stable Diffusion 3.5 (Esser et al., 2024) and FLUX.1-dev (Batifol et al., 2025), we discretize the continuous time interval $[0, 1]$ into a series of $N + 1$ timesteps $\{t_0, t_1, ..., t_N\}$, such that $t_i = \frac{\sigma i}{N + (\sigma - 1)i}$ with shift factor $\sigma = 3$.

**Inversion.** We first invert the origin image $x_{t_0}$ with null prompt $\varnothing$, preserving sufficient structural information for subsequent reconstruction. The discretized inversion trajectory is obtained by:

$$x_{t_{i+1}} = x_{t_i} + (t_{i+1} - t_i) \cdot \hat{v}_\theta(x_{t_i}, t_i, C_{inv}^p, C_{inv}^n), \tag{11}$$

where the factor $(t_{i+1} - t_i) > 0$ pushes the state towards noise. Meanwhile, the terms $\hat{v}_\theta(x_{t_i}, t_i, C_{rec}^p, C_{rec}^n)$ and $\frac{\partial}{\partial x_{t_i}} v_\theta(x_{t_i}, t_i, C_{rec}^p, C_{rec}^n)$ are also calculated, preparing the calculation of bypass $b_t^*$. Specifically, we set $C_{rec}^p = C_y$ (*i.e.*, target edit prompt) and $C_{rec}^n = C_x$ (*i.e.*, origin prompt) in the reconstruction trajectory. Importantly, our choice of prompts in FlowBypass is not an empirical hyperparameter decision but follows directly from the theoretical derivation of the bypass. The detail explanation will be given in Sec. 4.4.3.

The precise calculation of partial derivative $\frac{\partial}{\partial x_{t_i}} \hat{v}_\theta$ is computationally prohibitive for large backbones such as Stable Diffusion 3.5 or FLUX.1-dev. We approximate it via finite differences:

$$\frac{\partial}{\partial x_{t_i}} \hat{v}_\theta(x_{t_i}, t_i, C_y, C_x) \approx \frac{\hat{v}_\theta(x_{t_i} + \zeta, t_i, C_y, C_x) - \hat{v}_\theta(x_{t_i}, t_i, C_y, C_x)}{\zeta}, \tag{12}$$

where $\zeta$ is a small positive offset. This lightweight approximation introduces negligible bias while avoiding prohibitive memory and compute costs.

**Bypass Calculation.** We compute the bypass $b_t$ using the trapezoidal variant of Euler discretization applied to Equ. 10:

$$
\begin{aligned}
b_{t_i}^* &\approx -\frac{1}{2} \sum_{u=i}^{N-1} (t_{u+1} - t_u) \cdot (Q_u \cdot E_u + Q_{u+1} \cdot E_{u+1}), \\
Q_u &= \hat{v}_\theta(x_{t_u}, t_u, C_y, C_x) - \hat{v}_\theta(x_{t_u}, t_u, \varnothing, \varnothing), \\
E_u &= \Gamma\Big( -\frac{1}{2} \sum_{s=i}^{u-1} (t_{s+1} - t_s) \cdot (P_s + P_{s+1}) \Big), \\
P_s &= \frac{\hat{v}_\theta(x_{t_s} + \zeta, t_s, C_y, C_x) - \hat{v}_\theta(x_{t_s}, t_s, C_y, C_x)}{\zeta}, \\
\Gamma(x) &= \begin{cases} \exp(x), x \le 0 \\ x + 1, x > 0 \end{cases}.
\end{aligned}
\tag{13}
$$

To stabilize numerical evaluation, we replace the exponential with its first-order Taylor approximation in the positive domain, preventing uncontrolled growth.

**Reconstruction.** Unlike prior inversion–reconstruction pipelines, FlowBypass constructs the reconstruction trajectory starting from an intermediate state $y_{t_B}$, rather than from the inverted noise $y_{t_N}$. The reconstruction trajectory can be described as:

$$
\begin{aligned}
y_{t_B} &= x_{t_B} + b_{t_B}, \\
y_{t_{i-1}} &= y_{t_i} + (t_{i-1} - t_i) \cdot \hat{v}_\theta(y_{t_i}, t_i, C_y, C_x),
\end{aligned}
\tag{14}
$$

where $B$ is a user-specified bypass timestep between 0 and $N$, the factor $(t_{i-1} - t_i) < 0$ pushes the noisy state to the edit image $y_{t_0}$. A larger $B$ prioritizes alignment with edit prompt, while a smaller $B$ favors fidelity to the origin image. The solved $y_{t_0}$ corresponds to the desired edited image.

Table 1: Comparison with state-of-the-art image editing methods. OR denotes Optimization-Required, FM denotes Feature-Manipulation, SD is Stable Diffusion, LCM v7 is LCM Dreamshaper v7. Red, green, and blue highlight the best, second-best, and third-best results, respectively. The * marks indicate that the official code lacks real-image editing implementations, so we implement them ourselves following the authors' guidance.

| Method | Backbone | OR? | FM? | LPIPS↓ | I.Sim.↑ | T.Sim.↑ |
|---|---|---|---|---|---|---|
| P2P* (Hertz et al., 2023) | SD1.4 | No | Yes | 0.4990 | 81.04 | 26.93 |
| NTI (Mokady et al., 2023) | SD1.4 | Yes | Yes | 0.5798 | 73.96 | 26.38 |
| DDCM (Xu et al., 2024) | LCM v7 | No | Yes | 0.4507 | 87.14 | 26.62 |
| IP2P (Brooks et al., 2023) | SD1.4 | Yes | No | 0.6103 | 84.85 | 23.95 |
| Omni-Gen (Xiao et al., 2025) | Phi-3 | Yes | No | 0.3573 | 87.48 | 25.58 |
| LEDITS++ (Brack et al., 2024) | SD 1.5 | No | Yes | 0.3554 | 81.54 | 21.73 |
| RF-Solver (Wang et al., 2025) | FLUX.1-dev | No | Yes | 0.3880 | 87.32 | 25.30 |
| RF-Inversion (Rout et al., 2025) | FLUX.1-dev | No | No | 0.5659 | 83.35 | 25.71 |
| FluxSpace* (Dalva et al., 2025) | FLUX.1-dev | No | Yes | 0.8058 | 79.74 | 22.74 |
| FireFlow (Deng et al., 2025) | FLUX.1-dev | No | Yes | 0.3850 | 87.01 | 25.69 |
| FlowEdit (Kulikov et al., 2025) | FLUX.1-dev | No | No | 0.3921 | 87.90 | 25.28 |
| FlowBypass | SD3.5 Medium | No | No | 0.4228 | 85.96 | 26.45 |
| FlowBypass | SD3.5 Large | No | No | 0.4507 | 84.73 | 27.09 |
| FlowBypass | FLUX.1-dev | No | No | 0.3425 | 88.06 | 25.65 |

## 4 EXPERIMENTS

In this section, we present a comprehensive evaluation of our approach. Through both qualitative and quantitative analyses, we demonstrate the effectiveness and superiority of our approach. Furthermore, we conduct ablation studies to evaluate the contribution of each component in our design, including backbone robustness, approximation, and prompt choice, while the ablation studies of CFG scale, bypass step and hyperparameter $\zeta$ are provided in the Appendix.

### 4.1 EXPERIMENT SETUPS

**Baselines, Datasets, and Metrics.** We compare FlowBypass against a diverse set of state-of-the-art baselines, including DDIM-based approaches such as Null-Text Inversion (NTI) (Mokady et al., 2023), Denoising Diffusion Consistent Model (DDCM) (Xu et al., 2024), InstructPix2Pix (Brooks et al., 2023), Omni-Gen (Xiao et al., 2025), and LEDITS++ (Brack et al., 2024), as well as RF-based methods including RF-Solver (Wang et al., 2025) and RF-Inversion (Rout et al., 2025).

For evaluation, we adopt the EditEvalv2 benchmark (Huang et al., 2025), which contains 150 high-resolution images across seven sub-tasks: object addition, object replacement, object removal, background change, style change, texture change, and action change. Each image is resized while preserving its aspect ratio, with the longer side scaled to 1024 pixels for computational efficiency.

We assess editing performance from three complementary perspectives: (i) *perceptual fidelity*, measured by LPIPS (Zhang et al., 2018) between edited and original images; (ii) *semantic fidelity*, quantified by CLIP similarity (Radford et al., 2021) between edited and original images (*i.e.*, I.Sim.); and (iii) *semantic alignment*, evaluated by CLIPScore (Hessel et al., 2021) between edited images and edit prompts (*i.e.*, T.Sim.).

**Implementation Details.** FlowBypass is implemented in PyTorch and all experiments are conducted on a single NVIDIA GeForce RTX 4090 GPU. We evaluate on three widely used RF backbones: Stable Diffusion 3.5 Medium (*i.e.*, SD3.5M), Stable Diffusion 3.5 Large (*i.e.*, SD3.5L) (Esser et al., 2024), and FLUX.1-dev (*i.e.*, FLUX) (Batifol et al., 2025).

### 4.2 QUANTITATIVE COMPARISON WITH STATE-OF-THE-ART METHODS

We quantitatively compare FlowBypass with existing editing methods in Tab. 1. The * marks indicate that the official codebase does not provide implementations for real-image editing. Therefore,

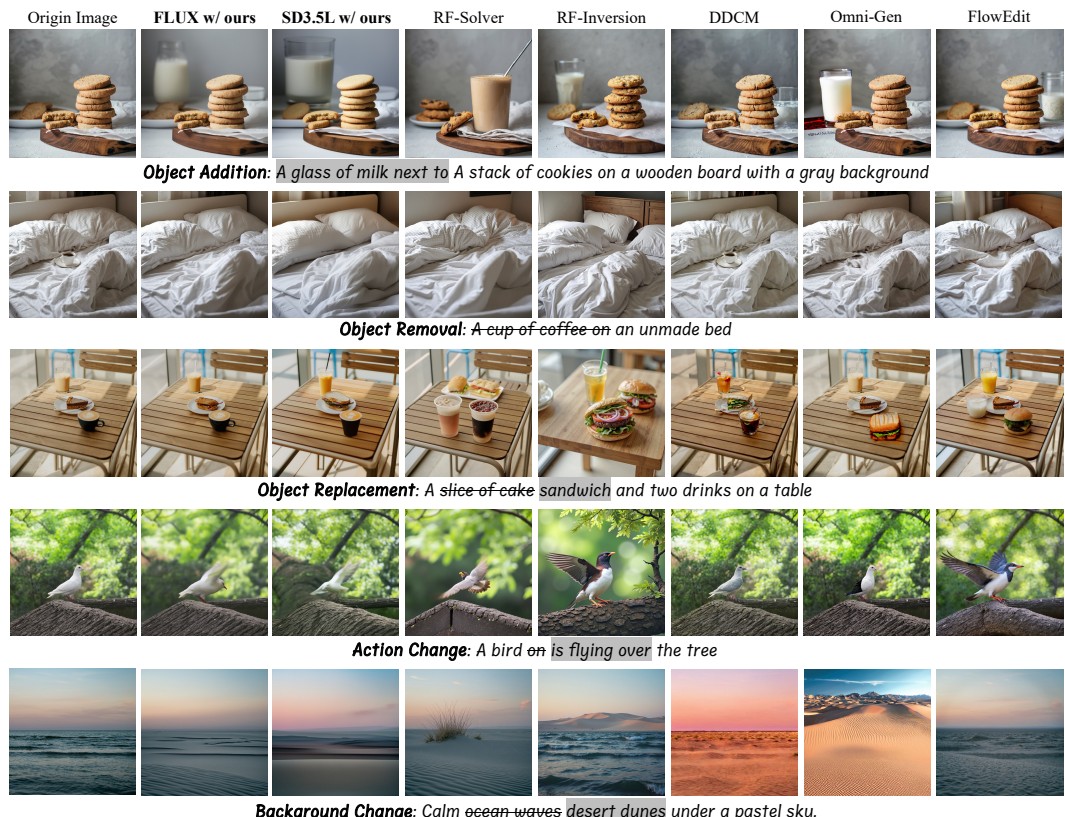

Figure 2: **Qualitative comparison with other image editing methods.** Zoom in for a better view.

we implement these components ourselves following the authors' guidelines. Importantly, superior editing performance cannot be judged by fidelity or alignment in isolation; an effective method must achieve a favorable balance across both dimensions. As reported in Tab. 1, FLUX w/ Flow-Bypass achieves the highest perceptual fidelity, while SD3.5L w/ FlowBypass demonstrates the best semantic alignment. More importantly, across all three backbones, FlowBypass consistently delivers strong alignment without substantially compromising fidelity. This trend is clearly reflected in Fig. 5, where the FlowBypass-related points are positioned closest to the top-left corner, approaching the Pareto frontier of fidelity and alignment. These results confirm the effectiveness of FlowBypass in achieving superior image editing performance with high fidelity and alignment.

### 4.3 QUALITATIVE COMPARISON WITH STATE-OF-THE-ART EDITING METHODS

The qualitative comparison with state-of-the-art image editing baselines is presented in Fig. 2. To assess the versatility of different approaches, we evaluate their performance across diverse sub-tasks. As shown, baseline methods often exhibit undesired modifications or incomplete edits, for example, unintentionally altering coffee cups in the Object Replacement sub-task or failing to animate the bird in the Action Change sub-task. In contrast, FlowBypass consistently delivers edits that closely follow the intended prompts while preserving structural and textural fidelity in regions that should remain untouched. This balance between alignment and fidelity underscores the robustness of Flow-Bypass and highlights its superior editing capability. More qualitative comparison and edited results outside the EditEvalv2 dataset can be obtained in the Appendix. We also conduct a user study to provide comprehensive subjective evidence of the visual performance of our method, which is discussed in the Appendix.

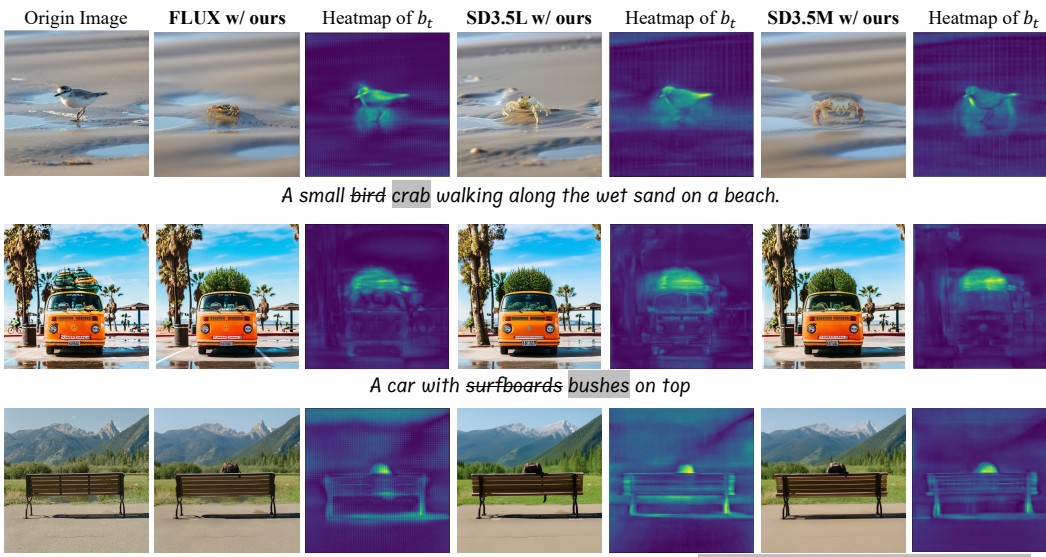

Figure 3: **Visualization of bypass.** Yellow regions indicate higher L1-norm values, while blue regions indicate lower values, reflecting the spatial distribution of bypass magnitude. Zoom in for a better view.

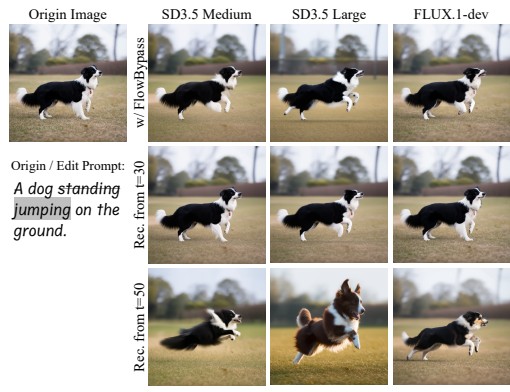

Figure 4: **Example outputs from different backbones and different settings.** Zoom in for a better view.

Table 2: Performance of different backbones.

| Backbone | Setting | LPIPS↓ | I.Sim.↑ | T.Sim.↑ |
|---|---|---|---|---|
| SD3.5M | w/ FlowBypass | 0.4228 | 85.96 | 26.45 |
| | Rec. from t=30 | 0.3082 | 90.67 | 24.87 |
| | Rec. from t=50 | 0.6354 | 77.18 | 27.57 |
| SD3.5L | w/ FlowBypass | 0.4507 | 84.73 | 27.09 |
| | Rec. from t=30 | 0.3288 | 91.01 | 25.36 |
| | Rec. from t=50 | 0.6576 | 76.95 | 27.94 |
| FLUX | w/ FlowBypass | 0.3425 | 88.06 | 25.65 |
| | Rec. from t=30 | 0.2240 | 94.47 | 23.85 |
| | Rec. from t=50 | 0.5811 | 78.21 | 27.78 |

## 4.4 ABLATION STUDY

### 4.4.1 ROBUSTNESS ON DIFFERENT BACKBONES

We perform an ablation study to evaluate the robustness of FlowBypass across three representative different backbones, including SD3.5M, SD3.5L, and FLUX. As mentioned before, the reconstruction trajectory starts from $y_{t_B} = x_{t_B} + b_{t_B}$. To verify the alignment contributed by FlowBypass, we conduct two diagnostic settings: (i) we set $b_{t_B} = 0$ and reconstruct from timestep $t = 30$, denoted as "Rec. from $t = 30$"; and (ii) we execute the entire reconstruction trajectory, starting from $t = 50$, to examine the fidelity from FlowBypass, denoted as "Rec. from $t = 50$".

As shown in Tab. 2, reconstructing the entire trajectory from $t = 50$ degrades fidelity due to error accumulation, while disabling the bypass at $t = 30$ prevents the framework from editing images according to the edit prompts. This observation is further supported by the visual examples in Fig. 4, where direct reconstruction from $t = 30$ yields under-edited outputs, whereas reconstruction from

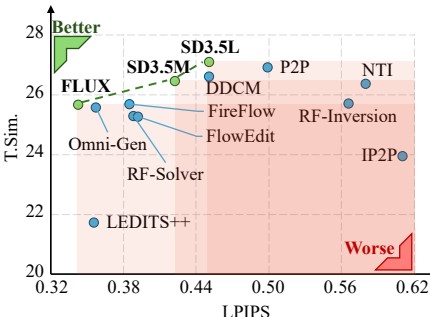

Figure 5: Scatter figure of quantitative comparison. Green points denote the performance of FlowBypass, whereas blue points denote the performance of other methods. FluxSpace is omitted because its extremely poor performance would compromise the readability of the scatter plot.

Table 3: Effectiveness of Approximation.

| Backbone | Setting | LPIPS↓ | I.Sim.↑ | T.Sim.↑ |
|---|---|---|---|---|
| SD3.5L | w/o FlowBypass | 0.3288 | 91.01 | 25.36 |
| | w/o Approx. $\frac{\partial}{\partial x_{t_i}}\hat{v}_\theta$ | 0.4035 | 86.67 | 26.73 |
| | w/o Approx. exp | 0.5719 | 78.12 | 26.22 |
| | w/ FlowBypass | 0.4507 | 84.73 | 27.09 |
| FLUX | w/o FlowBypass | 0.2240 | 94.47 | 23.85 |
| | w/o Approx. $\frac{\partial}{\partial x_{t_i}}\hat{v}_\theta$ | 0.3013 | 89.98 | 25.38 |
| | w/o Approx. exp | 0.4188 | 85.17 | 25.57 |
| | w/ FlowBypass | 0.3425 | 88.06 | 25.65 |

$t = 50$ introduces spurious changes, such as altering the dog's appearance. In contrast, our proposed FlowBypass introduces an appropriate bypass $b_{t_B}$, enabling faithful and precise editing that strikes a balance between fidelity and alignment across all backbones. The visualization in Fig. 3 further confirms that bypasses exhibit strong activations in the regions specified by the edit prompts, while remaining low in irrelevant areas that should stay consistent with the original images.

### 4.4.2 EFFECTIVENESS OF APPROXIMATION

We conduct an ablation study to evaluate the effectiveness of the approximations in Equ. 12 and Equ. 13. Since precisely computing the gradient is intractable, we set $\frac{\partial}{\partial x_{t_i}}\hat{v}_\theta$ in Equ. 12 to zero to assess the contribution of the gradient term in the bypass. In addition, we perform ablation experiments on the approximation of exp in Equ. 13 to examine its role in avoiding numerical instability. The results summarized in Tab. 3 indicate that approximating $\frac{\partial}{\partial x_{t_i}}\hat{v}_\theta$ leads to improved alignment, while introducing the approximation of exp further enhances both fidelity and alignment. We observe that neglecting the gradient approximation introduces unrealistic local structures, whereas omitting the approximation of exp leads to uncontrolled exponential growth, which results in severe artifacts in the edited images. More example outputs are provided in the Appendix.

### 4.4.3 IMPACT OF PROMPT CHOICE

The choice of prompts plays a crucial role in FlowBypass. As shown in Equ. 10, four prompts need to be determined: $C_{inv}^p$, $C_{inv}^n$, $C_{rec}^p$, and $C_{rec}^n$. To explore this design, we evaluate different reasonable prompt combinations, with results illustrated in Fig. 6. The detailed statistical results and qualitative comparison are provided in the Appendix.

Among them, the combination "ee/yx" which applies the empty prompt $\varnothing$ during inversion and uses $C_y$ as the positive prompt and $C_x$ as the negative prompt during reconstruction achieves the best balance between fidelity and alignment. We attribute this to two factors: (i) using the empty prompt $\varnothing$ in inversion preserves sufficient structural and semantic information from the origin images in the inversion trajectory, thus benefiting fidelity; and (ii) the balanced non-linear compensation constructed by $C_{rec}^p = C_y, C_{rec}^n = C_x$. Specifically, in the analytical solution of a first-order linear differential equation, the exponential term acts as the accumulated effect of nonlinear terms along the trajectory. In FlowBypass, this exponential term compensates for the semantic discrepancies introduced by the Taylor expansion. Consequently, using the "yx" prompt configuration provides a balanced compensation for this integrating term and amplifies the semantic shift from origin semantics to edit semantics. Alternative settings (e.g., "ye" and "yy") would create an imbalanced compensation, causing the bypass computation to overly favor either the target or the origin image, thereby harming fidelity or alignment.

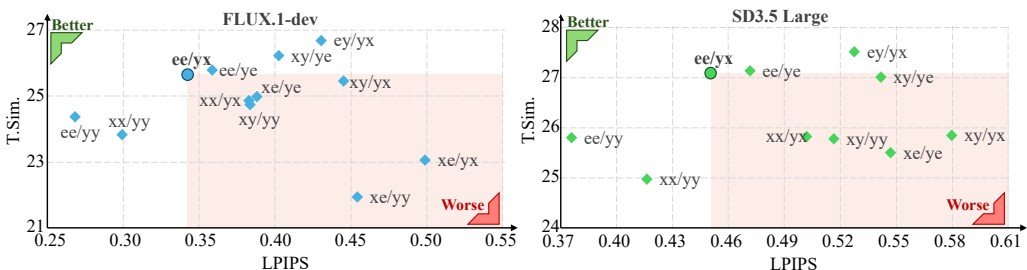

Figure 6: **Performance of different prompt choices.** In each notation, the segment before the slash denotes inversion prompts and the segment after the slash denotes reconstruction prompts. Within each segment, the first character indicates the positive prompt and the second indicates the negative prompt, where "x" denotes $C_x$, "y" denotes $C_y$, and "e" denotes $\varnothing$.

## 5 LIMITATION

Despite its effectiveness, FlowBypass still presents several limitations that highlight opportunities for future improvement. First, FlowBypass do not show its superiority on editing speed, as computing the bypass term $b_t$ imposes additional overhead. Although FlowBypass is not the fastest method in absolute runtime, it achieves a practical processing cost for high-resolution image editing while offering superior performance and a well balance between fidelity and alignment. Second, FlowBypass shows limited reliability on negation-based prompts (e.g., "without"). Such edits may fail because the backbone generative backbones inherently struggle with negative conditioning. For instance, prompts such as "a cat without a hat" often still produce a cat wearing a hat. This issue is shared by many editing methods built upon these backbones, as the models themselves provide weak and unreliable outputs for negation. Reformulating negation into affirmative phrasing (e.g., "a cat with a hat" → "a cat") yields more stable and reasonable results.

## 6 CONCLUSION

In this work, we propose FlowBypass, a general rectified-flow image editing framework that achieves high alignment with target edit prompts while preserving consistency with irrelevant regions of the original images. It is accomplished by introducing a bypass between the inversion and reconstruction trajectories, without requiring any additional training, test-time optimization, or feature manipulation. The framework is theoretically motivated by the formulation of bypassing across two trajectories and is realized through carefully designed approximations and prompt selection strategies. Extensive experiments demonstrate that FlowBypass outperforms existing state-of-the-art methods, striking a superior balance between fidelity and alignment.

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

## A APPENDIX

### A.1 SOLVING OF ODE

The ODE described in Equ. 9 can be solved using the Integrating Factor Method. Equ. 9 can be rewritten as:

$$
\begin{aligned}
\frac{d}{dt} b_t^* &= Q(t) + P(t) \cdot b_t^*, \\
Q(t) &= \hat{v}_\theta(x_t, t, C_{rec}^p, C_{rec}^n) - \hat{v}_\theta(x_t, t, C_{inv}^p, C_{inv}^n), \\
P(t) &= \frac{\partial}{\partial x_t} \hat{v}_\theta(x_t, t, C_{rec}^p, C_{rec}^n), \\
b_1^* &= 0.
\end{aligned}
\tag{15}
$$

Let

$$
\mu(t) = \exp\left(-\int_1^t P(s)ds\right),
\tag{16}
$$

and it can be obtained that

$$
\frac{d}{dt}[\mu(t)b_t^*] = \mu(t)\frac{d}{dt}b_t^* - \mu(t)P(t)b_t^* = \mu(t)Q(t).
\tag{17}
$$

Integrating both sides of the equation with respect to $t$ from 1 yields:

$$
\mu(t)b_t^* - \mu(1)b_1^* = \int_1^t \mu(u)Q(u)du.
\tag{18}
$$

Due to $b_1^* = 0$ and $\mu(t) > 0$ for any $t$, Equ. 18 can be rearranged as:

$$
\begin{aligned}
b_t^* &= \frac{\int_1^t \mu(u)Q(u)du + 0}{\mu(t)} \\
&= \exp\left(\int_1^t P(s)ds\right) \int_1^t \exp\left(-\int_1^u P(s)ds\right)Q(u)du \\
&= \int_1^t \exp\left(\int_1^t P(s)ds - \int_1^u P(s)ds\right)Q(u)du \\
&= \int_1^t \exp\left(\int_u^t P(s)ds\right)Q(u)du \\
&= \int_1^t \left(\hat{v}_\theta(x_u, u, C_{rec}^p, C_{rec}^n) - \hat{v}_\theta(x_u, u, C_{inv}^p, C_{inv}^n)\right) \exp\left(\int_u^t \frac{\partial}{\partial x_s}\hat{v}_\theta(x_s, s, C_{rec}^p, C_{rec}^n)ds\right)du,
\end{aligned}
\tag{19}
$$

which is equal to Equ. 10.

### A.2 ADDITIONAL EXPERIMENT RESULTS

#### A.2.1 DETAILED STATISTICAL ANALYSIS OF ABLATION ON PROMPT CHOICE

The detailed statistical results of Fig. 6 in main text is demonstrated in Tab. 4. Besides, we provide the visual comparison with different prompt choices in Fig. 7, which indicates that the combination "ee/yx" preserves the fidelity of irrelevant regions while successfully applying the intended edit.

#### A.2.2 VISUALIZATION OF APPROXIMATION

Fig. 8 shows the ablation results on approximation. Severe artifacts appear when the approximation of exp is omitted, as illustrated in the fourth and seventh columns. We argue that these artifacts

Table 4: Impact of Prompt Choice.

| Backbone | | FLUX.1-dev | | | SD3.5 Large | | |
|---|---|---|---|---|---|---|---|
| Inverse | Edit | LPIPS↓ | I.Sim.↑ | T.Sim.↑ | LPIPS↓ | I.Sim.↑ | T.Sim.↑ |
| $(+C_x, -C_y)$ | $(+C_y, -C_x)$ | 0.4454 | 83.41 | 25.47 | 0.5798 | 77.58 | 25.85 |
| $(+C_x, -C_y)$ | $(+C_y, -\varnothing)$ | 0.4024 | 86.07 | 26.23 | 0.5418 | 81.25 | 27.01 |
| $(+C_x, -C_y)$ | $(+C_y, -C_y)$ | 0.3836 | 86.54 | 24.74 | 0.5166 | 81.04 | 25.78 |
| $(+C_x, -\varnothing)$ | $(+C_y, -C_x)$ | 0.4990 | 79.11 | 23.07 | 0.7083 | 68.29 | 20.01 |
| $(+C_x, -\varnothing)$ | $(+C_y, -\varnothing)$ | 0.3881 | 86.19 | 24.99 | 0.5471 | 79.72 | 25.50 |
| $(+C_x, -\varnothing)$ | $(+C_y, -C_y)$ | 0.4542 | 82.28 | 21.95 | 0.6834 | 69.48 | 18.79 |
| $(+C_x, -C_x)$ | $(+C_y, -C_y)$ | 0.2992 | 90.85 | 23.84 | 0.4163 | 87.15 | 24.98 |
| $(+C_x, -C_x)$ | $(+C_y, -C_x)$ | 0.3828 | 86.10 | 24.87 | 0.5022 | 81.40 | 25.83 |
| $(+\varnothing, -C_y)$ | $(+C_y, -C_x)$ | 0.4305 | 85.66 | 26.69 | 0.5277 | 82.46 | 27.52 |
| $(+\varnothing, -\varnothing)$ | $(+C_y, -C_y)$ | 0.2685 | 93.16 | 24.38 | 0.3759 | 90.00 | 25.80 |
| $(+\varnothing, -\varnothing)$ | $(+C_y, -\varnothing)$ | 0.3586 | 89.65 | 25.80 | 0.4716 | 85.55 | 27.14 |
| $(+\varnothing, -\varnothing)$ | $(+C_y, -C_x)$ | **0.3425** | **88.06** | **25.65** | **0.4507** | **84.73** | **27.09** |

Table 5: Impact of CFG scale.

| Backbone | FLUX.1-dev | | | SD3.5 Large | | |
|---|---|---|---|---|---|---|
| CFG scale | LPIPS↓ | I.Sim.↑ | T.Sim.↑ | LPIPS↓ | I.Sim.↑ | T.Sim.↑ |
| 1.5 | 0.3036 | 90.55 | 25.23 | 0.4155 | 87.04 | 26.83 |
| 2.0 | **0.3425** | **88.06** | **25.65** | **0.4507** | **84.73** | **27.09** |
| 2.5 | 0.3731 | 86.55 | 26.14 | 0.4848 | 82.51 | 27.23 |
| 3.0 | 0.4021 | 85.49 | 26.30 | 0.5160 | 81.40 | 27.06 |
| 3.5 | 0.4350 | 84.13 | 26.28 | 0.5472 | 79.84 | 26.74 |
| 4.0 | 0.4642 | 83.18 | 26.17 | 0.5773 | 78.56 | 26.46 |
| 4.5 | 0.4940 | 81.69 | 26.14 | 0.6027 | 77.24 | 26.34 |

mainly arise from exponential explosion, which injects excessively large values into the calculated bypass, causing the normalization layers in the denoiser network to behave improperly, and finally introduces these unpleasant artifacts. Furthermore, removing the approximation of $\frac{\partial}{\partial x_{t_i}}\hat{v}_\theta$ introduces unrealistic structural details, such as distorted stones in the first row, deformed bicycles in the second row, and abnormal house shapes in the third row. These visualizations indicate that the introduced approximations not only produce more realistic details but also effectively suppress artifacts that may arise from exponential explosion.

## A.3 MORE EXPERIMENT RESULTS

### A.3.1 MORE QUALITATIVE COMPARISON WITH SOTA EDITING METHODS

Additional qualitative comparisons are provided in Fig. 11. The results demonstrate that FlowBypass, across different backbones, outperforms other editing methods in terms of both fidelity and alignment.

### A.3.2 USER STUDY

A user study is conducted to provide comprehensive subjective evidence of the visual performance of our method. Participants were asked to compare our results with those produced by state-of-the-art approaches in a pairwise preference setting under controlled viewing conditions. As shown in Fig. 12, our method consistently achieves the highest preference rates across all backbones and comparison methods. In each bar, the green region indicates the proportion of users who preferred our results, while the red region corresponds to the competing method. These results demonstrate that FlowBypass not only improves objective reconstruction fidelity but also delivers outputs that align more closely with human perception.

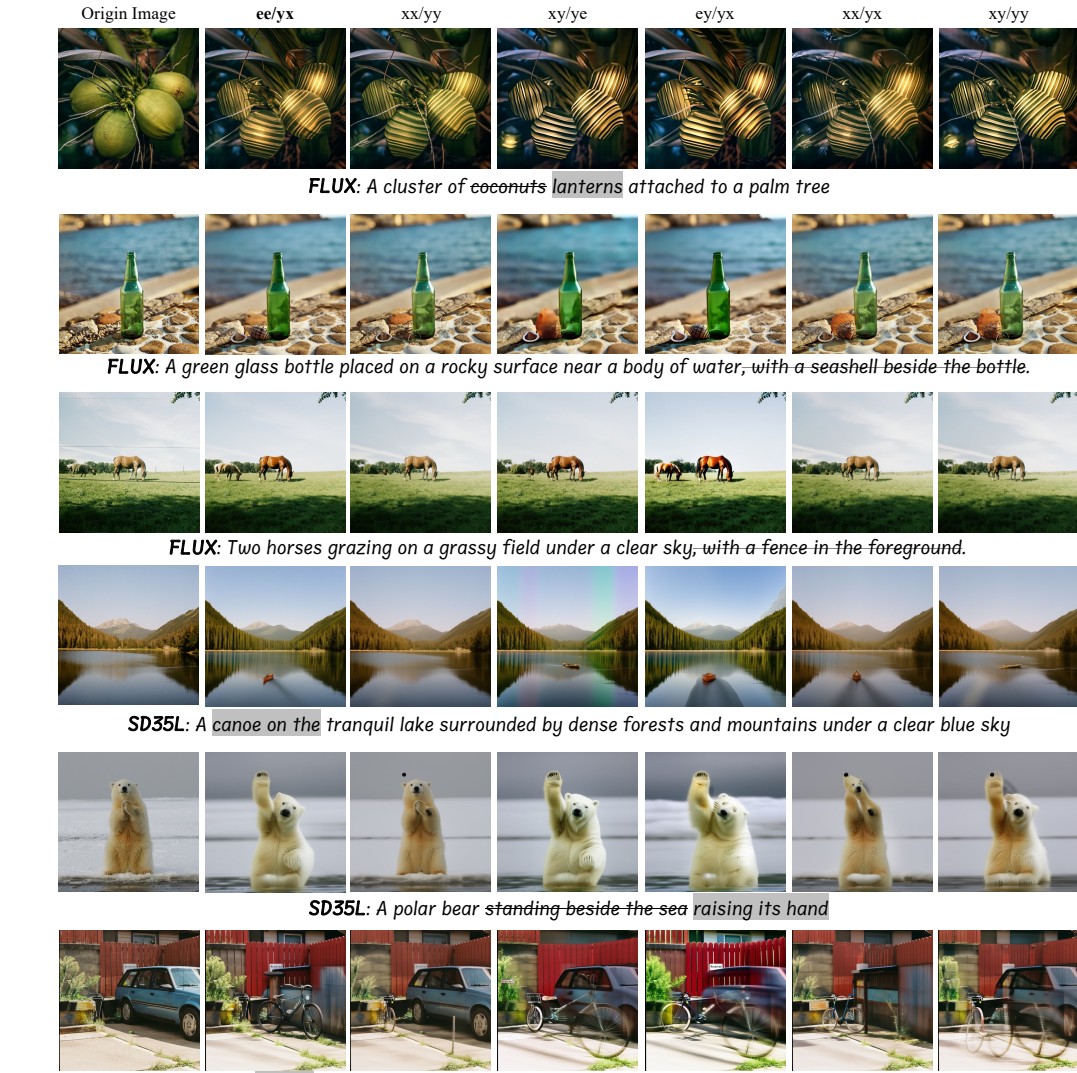

|  | Origin Image | ee/yx | xx/yy | xy/ye | ey/yx | xx/yx | xy/yy |

*FLUX*: A cluster of ~~coconuts~~ lanterns attached to a palm tree

*FLUX*: A green glass bottle placed on a rocky surface near a body of water, ~~with a seashell beside the bottle.~~

*FLUX*: Two horses grazing on a grassy field under a clear sky, ~~with a fence in the foreground.~~

*SD35L*: A canoe on the tranquil lake surrounded by dense forests and mountains under a clear blue sky

*SD35L*: A polar bear ~~standing beside the sea~~ raising its hand

*SD35L*: An old blue ~~car~~ bicycle parked in a driveway next to a red wooden fence, with a 'Reserved' sign in the background.

Figure 7: **Visual comparison of ablation results on different prompt choices.** Zoom in for a better view.

Table 6: Impact of bypass step.

| Backbone | FLUX.1-dev | | | SD3.5 Large | | |
|---|---|---|---|---|---|---|
| Timestep | LPIPS↓ | I.Sim.↑ | T.Sim.↑ | LPIPS↓ | I.Sim.↑ | T.Sim.↑ |
| 10 | 0.1849 | 96.14 | 23.36 | 0.3003 | 92.84 | 24.50 |
| 20 | 0.2525 | 92.99 | 24.66 | 0.3695 | 89.37 | 25.86 |
| 30 | **0.3425** | **88.06** | **25.65** | **0.4507** | **84.73** | **27.09** |
| 40 | 0.4490 | 84.25 | 27.16 | 0.5618 | 80.20 | 27.70 |
| 50 | 0.5811 | 78.21 | 27.78 | 0.6576 | 76.95 | 27.94 |

### A.3.3 RUNTIME COMPARISON WITH SOTA EDITING METHODS

We present a comprehensive runtime comparison against state-of-the-art editing methods in Tab. 7. As specified in Sec. 4.1, all experiments are conducted on a single RTX 4090 GPU. However, FLUX.1-dev cannot be executed naively under the 24GB VRAM limit of the RTX 4090. To en-

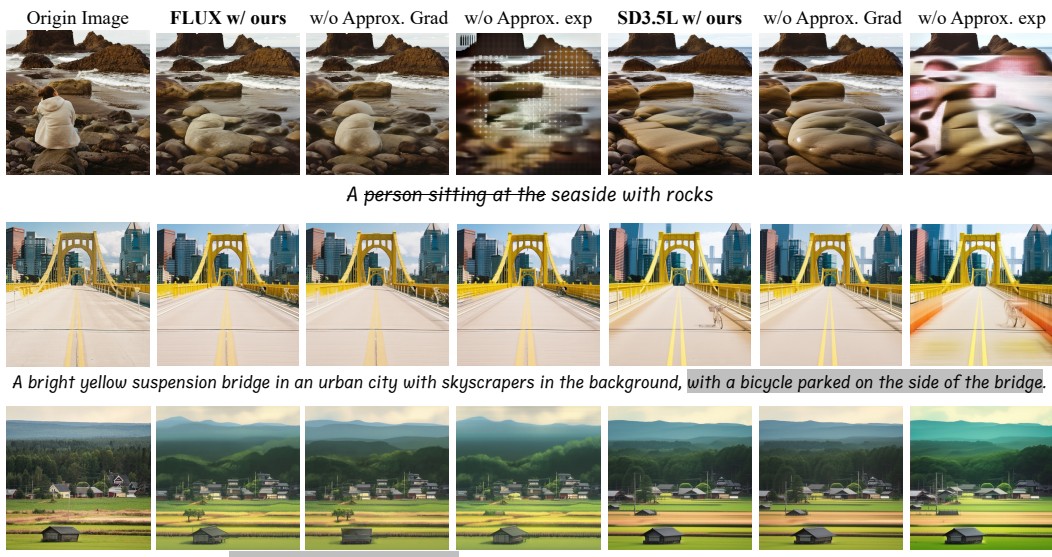

| Origin Image | **FLUX w/ ours** | w/o Approx. Grad | w/o Approx. exp | **SD3.5L w/ ours** | w/o Approx. Grad | w/o Approx. exp |

*A ~~person sitting at~~ the seaside with rocks*

*A bright yellow suspension bridge in an urban city with skyscrapers in the background, with a bicycle parked on the side of the bridge.*

*A Japanese anime style of A countryside field with houses and trees*

Figure 8: **Visual comparison of ablation results on approximation.** Zoom in for a better view.

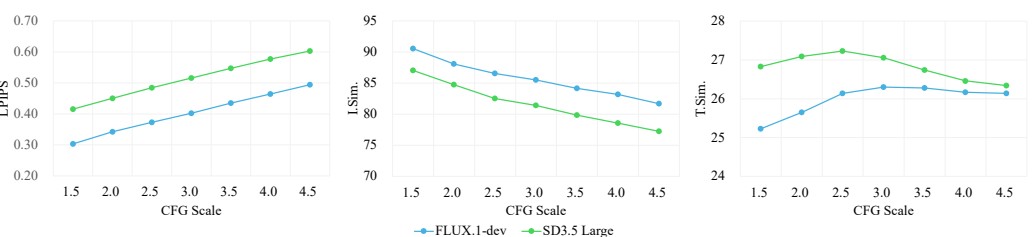

Figure 9: Trends of edit performance as CFG scale $\omega$ increases.

sure functional equivalence while fitting the model into memory, we employ DeepSpeed with ZeRO Stage-3 optimization Rajbhandari et al. (2020), which preserves computational correctness while reducing memory consumption. It is worth noting that the runtime of FLUX on RTX 4090 is sensitive to system-level factors such as PCIe bandwidth and host RAM capacity. To provide a fairer comparison, we additionally benchmark all methods on a larger-memory GPU (L20) without Deep-Speed, as reported in the fifth column of Tab. 7. Although FlowBypass is not the fastest method in absolute runtime, it achieves a practical processing cost for high-resolution image editing while offering superior performance and a well balance between fidelity and alignment.

Additionally, we break down the computational cost into different stages to highlight the extra overhead introduced by the bypass, as shown in Tab. 8. Theoretically, the bypass computation should account for only about 21.1% additional cost, and our practical results confirm this, demonstrating that the bypass is not the primary contributor to the overall computational burden. The discrepancy between the totals reported in Tab. 8 and Tab. 7 comes from the exclusion of text encoding time in Tab. 8.

### A.3.4 MORE EDITED RESULTS OUTSIDE DATASET

To further evaluate the generalization ability of FlowBypass, we conduct image editing experiments on samples outside the EditEvalv2 dataset, as shown in Fig. 13. The origin images are selected from four sources, namely the impressionist painting of Claude Monet (first of row 2), the natural image dataset Flickr2K (Timofte & Agustsson, 2017), the image generated by SD3.5L (second of row 4),

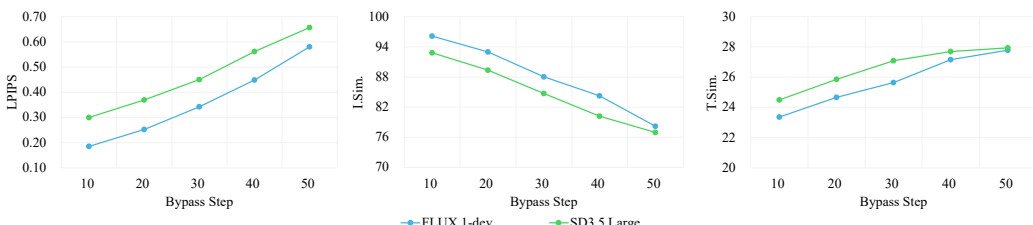

Figure 10: Trends of edit performance as Bypass step $t_B$ changes.

Table 7: Runtime comparison with SOTA editing methods at $1024 \times 1024$ resolution. The # in parentheses indicates the ranking.

| Method | Backbone | Precision | 4090 Runtime (s)↓ | L20 Runtime (s)↓ |
|---|---|---|---|---|
| P2P* | SD1.4 | FP32 | 17.59 (#3) | 24.41 (#4) |
| NTI | SD1.4 | FP32 | 424.03 (#14) | 651.65 (#14) |
| DDCM | LCM v7 | FP16 | 5.36 (#1) | 6.47 (#1) |
| IP2P | SD1.4 | FP32 | 8.28 (#2) | 22.16 (#3) |
| Omni-Gen | Phi-3 | FP32 | 81.59 (#8) | 101.84 (#11) |
| LEDITS++ | SD 1.5 | FP32 | 33.19 (#6) | 45.18 (#7) |
| RF-Solver | FLUX.1-dev | BF16 | 256.23 (#13) | 138.33 (#12) |
| RF-Inversion | FLUX.1-dev | BF16 | 117.24 (#10) | 51.78 (#8) |
| FluxSpace* | FLUX.1-dev | BF16 | 145.50 (#11) | 65.75 (#9) |
| FireFlow | FLUX.1-dev | BF16 | 29.85 (#5) | 20.66 (#2) |
| FlowEdit | FLUX.1-dev | BF16 | 101.08 (#9) | 43.95 (#6) |
| FlowBypass | SD3.5 Medium | FP16 | 29.43 (#4) | 35.90 (#5) |
| FlowBypass | SD3.5 Large | FP16 | 64.11 (#7) | 81.31 (#10) |
| FlowBypass | FLUX.1-dev | BF16 | 172.24 (#12) | 192.23 (#13) |

and Pexels online website[1] (second and third of row 6). The results indicate that FlowBypass can effectively edit diverse types of images with fidelity and alignment, which demonstrates its general editing capability. We also observe that the fidelity is slightly lower when editing the SD3.5L-generated image compared with natural images, as shown in the last of fourth row. This difference may be caused by the domain gap of generation characteristics between FLUX and SD3.5L. In general, FlowBypass can still handle images that follows the unseen distribution during training to provide outputs with desired editing and fidelity to the original image.

### A.3.5 IMPACT OF CFG SCALE

We conduct ablation experiments on the CFG scale $\omega$ to examine the effect of different guidance strengths on editing performance, with results presented in Fig. 9. As the CFG scale increases, fidelity gradually decreases, while alignment first improves and then declines. When the CFG scale becomes excessively large, overly strong guidance introduces noticeable artifacts and abnormal appearances. Fig. 14 illustrates this trend, showing that smaller CFG scales preserve fidelity but limit alignment, whereas larger CFG scales behave more aggressively and may even introduce severe artifacts. Based on these findings, we set the CFG scale $\omega$ to 2 in our experiments. The detailed statistical results are provided in Tab. 5.

### A.3.6 IMPACT OF HYPERPARAMETER $\zeta$

We perform ablation study on the hyperparameter $\zeta$ in Equ. 12. The results in Tab. 9 indicate that FlowBypass is reasonably robust to $\zeta$, and varying it within a practical range does not meaningfully influence editing quality, which partially validates the assumption underlying the Taylor expansion in Equ. 12.

---

[1][Online] Available at https://www.pexels.com

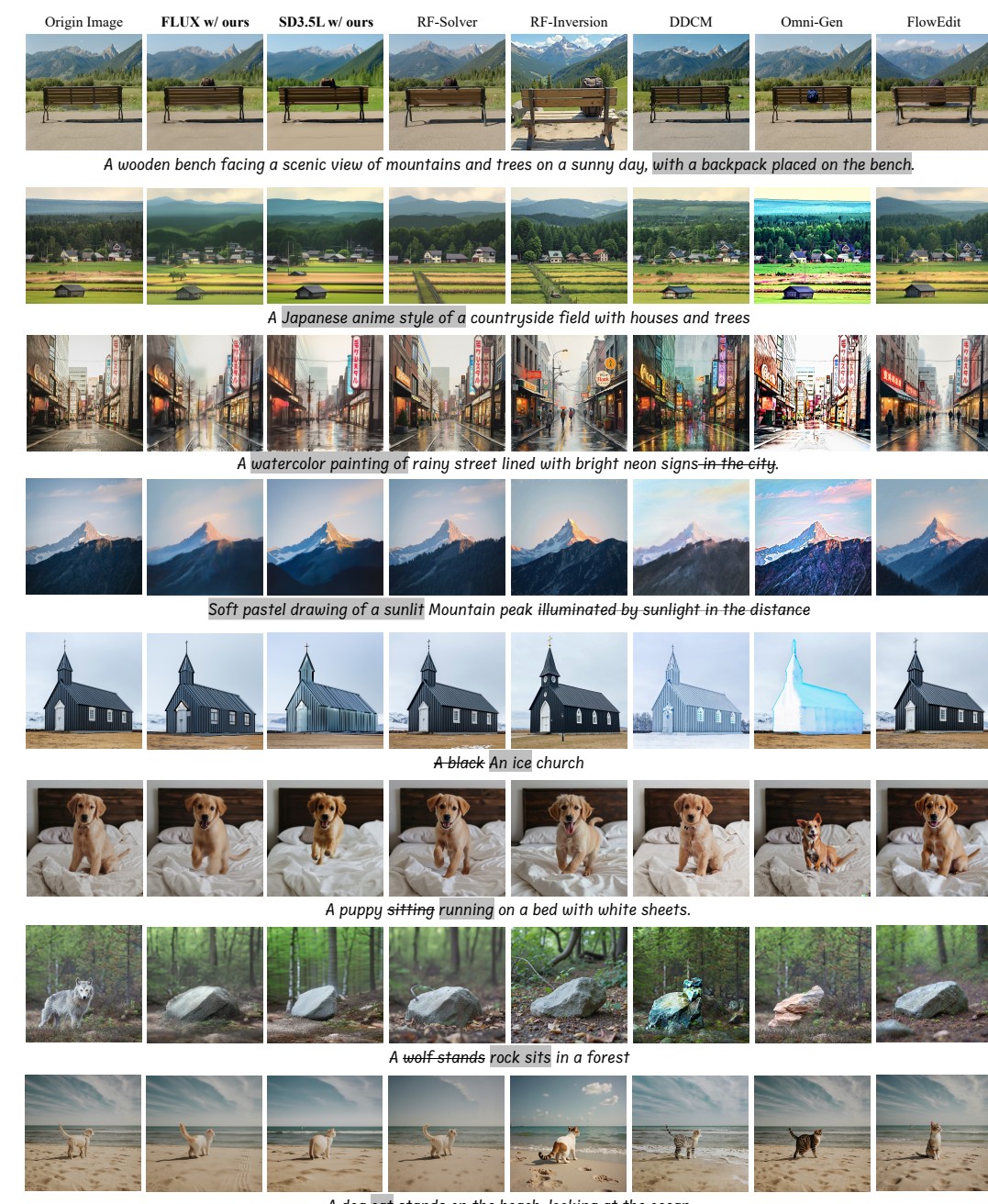

Figure 11: **More qualitative comparison with SOTA editing methods.** Zoom in for a better view.

### A.3.7 IMPACT OF BYPASS STEP

We conduct an ablation study about the impact of bypass step $t_B$, whose trends are presented in Fig. 10. The results reveal a clear trade-off between fidelity and alignment, where fidelity decreases monotonically and alignment increases monotonically as $t_B$ increases, as demonstrated in Fig. 15. We choose $t_B = 30$ for the best balance between fidelity and alignment. The detailed statistical results are provided in Tab. 6.

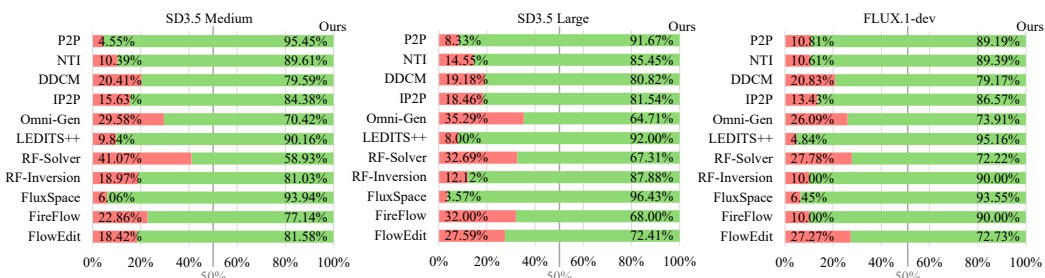

Figure 12: User Study Statistics. The green portion represents the percentage of users who rated our method's output as better, while the red portion represents the percentage who preferred the output of the compared method.

Table 8: Computational cost breakdown (second). The % in parentheses indicates the percentage.

| Stage | SD35M | SD35L | FLUX.1 |
|---|---|---|---|
| VAE encode | 0.23 (1.08%) | 0.16 (0.31%) | 0.22 (0.13%) |
| Inversion | 9.71 (46.25%) | 24.38 (46.97%) | 78.50 (46.52%) |
| **Inversion for Bypass** | 4.32 (20.56%) | 10.84 (20.87%) | 34.89 (20.67%) |
| **Bypass** | 0.00836 (0.0398%) | 0.00335 (0.006461%) | 0.00359 (0.002129%) |
| Recon | 6.42 (30.56%) | 16.27 (31.34%) | 54.90 (32.53%) |
| VAE decode | 0.32 (1.51%) | 0.26 (0.50%) | 0.24 (0.14%) |

### A.3.8 ABLATION STUDY ON DIFFERENT RECONSTRUCTION TIMESTEP WITHOUT BYPASS

We conduct an ablation study to evaluate the performance of inversion–reconstruction editing paradigm when no bypass is introduced, and the reconstruction starts not from pure noise but from an intermediate latent state. As shown in Tab. 10, when starting from larger $t$, fidelity is difficult to maintain, whereas starting from smaller $t$ compromises alignment. This phenomenon provides an evidence from opposite that FlowBypass addresses the trade-off by correctly jumping the reconstruction starting point onto the reconstruction trajectory, achieving a balance between fidelity and alignment and yielding more stable, high-quality edits.

### A.3.9 VISUALIZATION OF BYPASS $b_t$ UNDER DIFFERENT $t$

We present a series of visualization of bypass $b_t$ under different $t$ during editing. We would like to clarify that FlowBypass performs only one bypass computation and transition during an actual editing process. This visualization is performed solely to illustrate how the bypass behaves when applied at different bypass timesteps. As illustrated in Fig. 16, larger values of $t_B$ (*i.e.*, earlier denoising stages) tend to influence global layout and structure, while smaller values of $t_B$ (*i.e.*, later denoising stages) exhibit the modification of local details and texture refinement. This pattern aligns well with observations reported in prior works on DDIM and RF-based sampling, and provides an intuitive view of how the bypass mechanism modulates semantic corrections across the trajectories.

### A.4 ETHICS STATEMENT

This work focuses on developing image editing methods that aim to achieve high alignment with edit prompts and fidelity in irrelevant regions. We recognize that image editing technologies carry potential risks, including manipulating content that could mislead viewers, spread misinformation, or infringe upon individual privacy and rights. Therefore, we make effort to enhance the recognizability of edited images, including applying an invisible watermark and recording edit information in the image file metadata. Our research is conducted under the principle of responsible innovation. Our experiments do not involve sensitive personal data, private information, or identifiable individuals. All images used in our evaluation are either synthetic or drawn from publicly available datasets that permit research use. We emphasize that our contribution should be applied in lawful and ethical

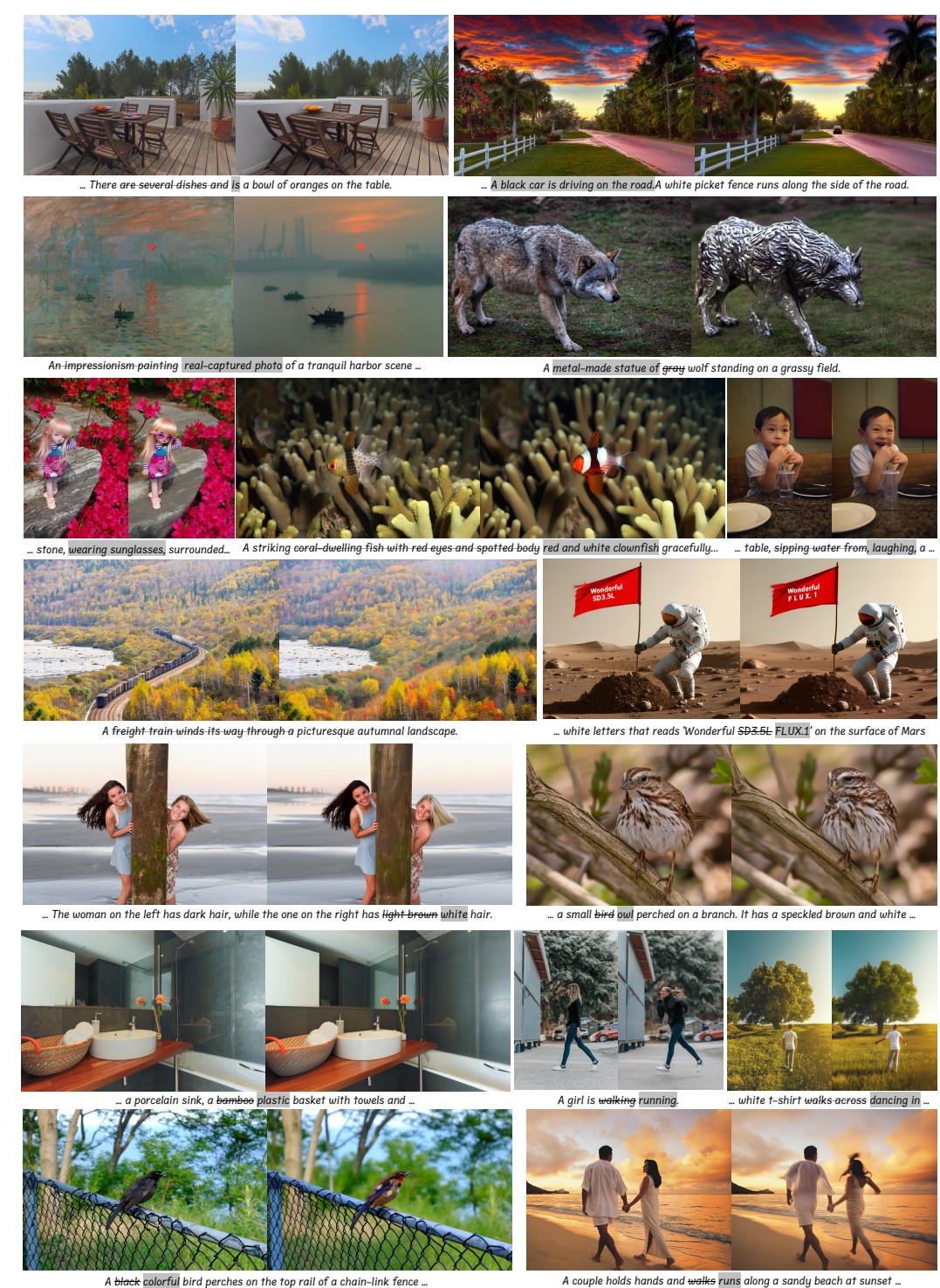

Figure 13: **Edit results outside dataset.** Zoom in for a better view.

contexts, and we caution against its use for creating harmful or deceptive images. By improving the controllability and reliability of image editing models, we aim to contribute positively to the safe and trustworthy deployment of image editing technologies.

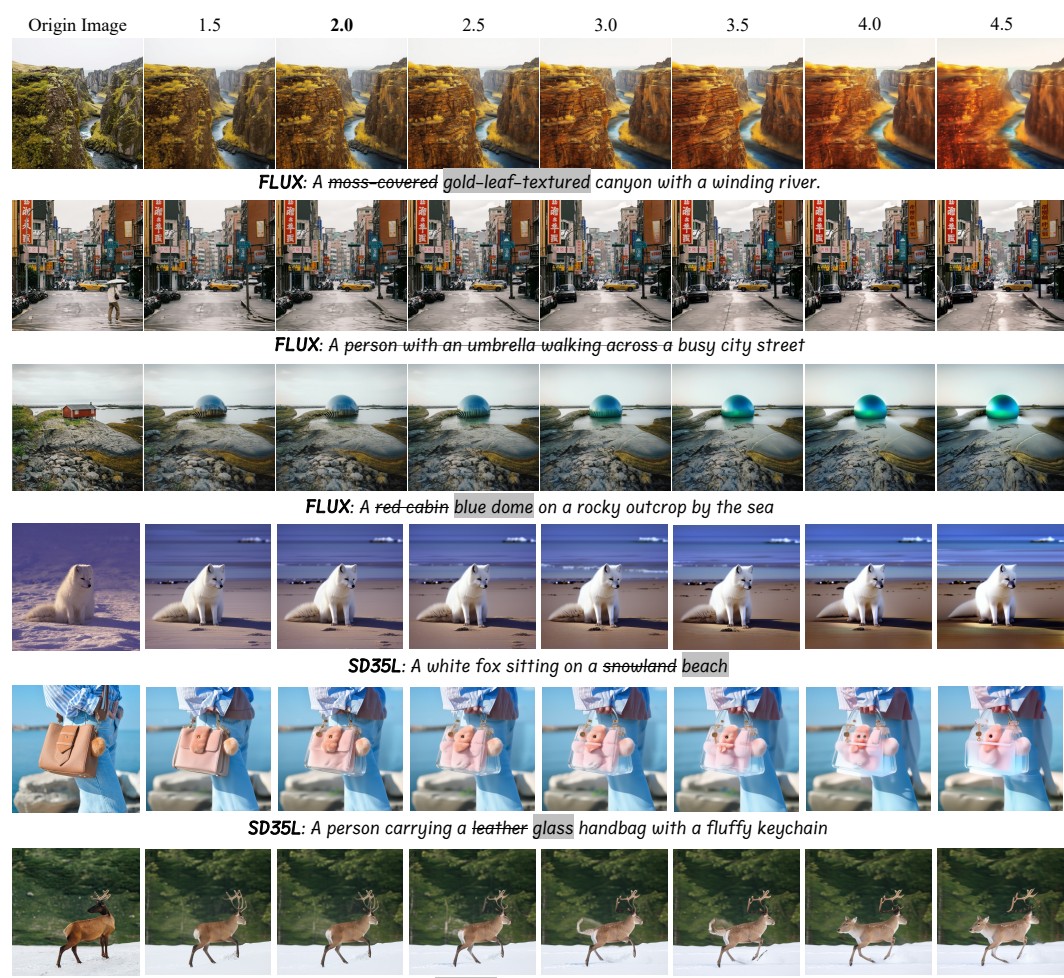

Figure 14: **Visual comparison of ablation results on different CFG scales.** Zoom in for a better view.

Table 9: Impact of hyperparameter $\zeta$.

| Backbone | FLUX.1-dev | | | SD3.5 Large | | |
|---|---|---|---|---|---|---|
| $\zeta$ | LPIPS↓ | I.Sim.↑ | T.Sim.↑ | LPIPS↓ | I.Sim.↑ | T.Sim.↑ |
| 0.001 | 0.3290 | 88.56 | 25.66 | 0.4513 | 84.77 | 27.10 |
| 0.005 | 0.3446 | 87.91 | 25.77 | 0.4488 | 85.09 | 27.14 |
| 0.01 | **0.3425** | **88.06** | **25.65** | **0.4507** | **84.73** | **27.09** |
| 0.05 | 0.3509 | 88.03 | 25.84 | 0.4606 | 84.65 | 27.21 |
| 0.1 | 0.3434 | 88.25 | 25.82 | 0.4562 | 84.81 | 27.32 |

## A.5 LLM USAGE STATEMENT

We affirm that large language models (LLMs) were not used at any stage of this work, including research ideation, experimental design, data analysis, or manuscript writing. All ideas, methods, experiments, and results were conceived and conducted entirely by the authors, who take complete responsibility for the content of this paper.

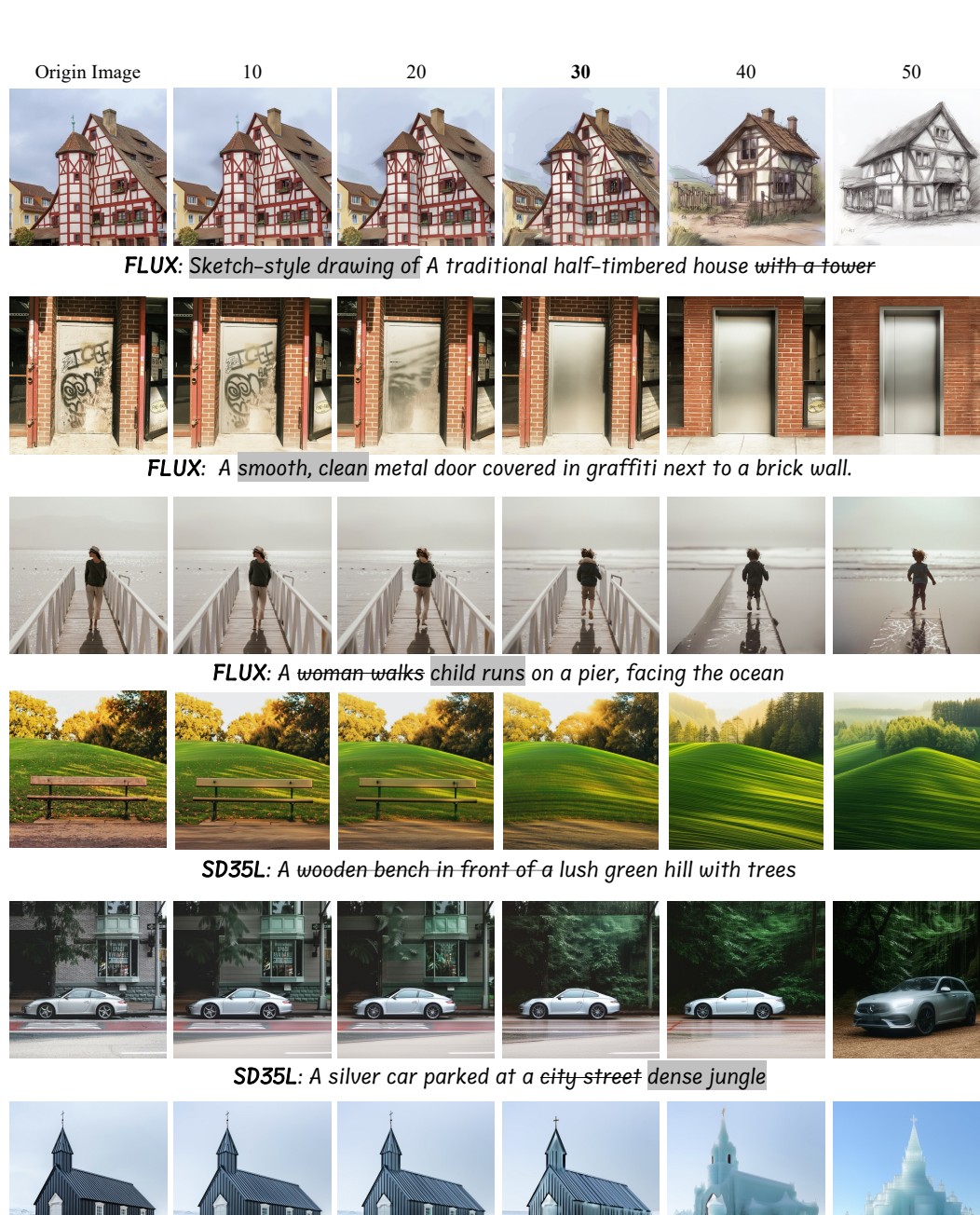

Figure 15: **Visual comparison of ablation results on different bypass steps.** Zoom in for a better view.

Table 10: Ablation study on different reconstruction timestep without bypass.

| Backbone | FLUX.1-dev | | | SD3.5 Large | | |
|---|---|---|---|---|---|---|
| Rec. t | LPIPS↓ | I.Sim.↑ | T.Sim.↑ | LPIPS↓ | I.Sim.↑ | T.Sim.↑ |
| 10 | 0.1585 | 97.56 | 22.35 | 0.2561 | 95.50 | 22.81 |
| 20 | 0.1834 | 96.94 | 22.88 | 0.2817 | 94.31 | 23.82 |
| 30 | 0.2240 | 94.47 | 23.85 | 0.3288 | 91.01 | 25.36 |
| 40 | 0.3358 | 88.32 | 25.53 | 0.4487 | 84.43 | 26.72 |
| 50 | 0.5811 | 78.21 | 27.78 | 0.6576 | 76.95 | 27.94 |
| 30 w/ Bypass | **0.3425** | **88.06** | **25.65** | **0.4507** | **84.73** | **27.09** |

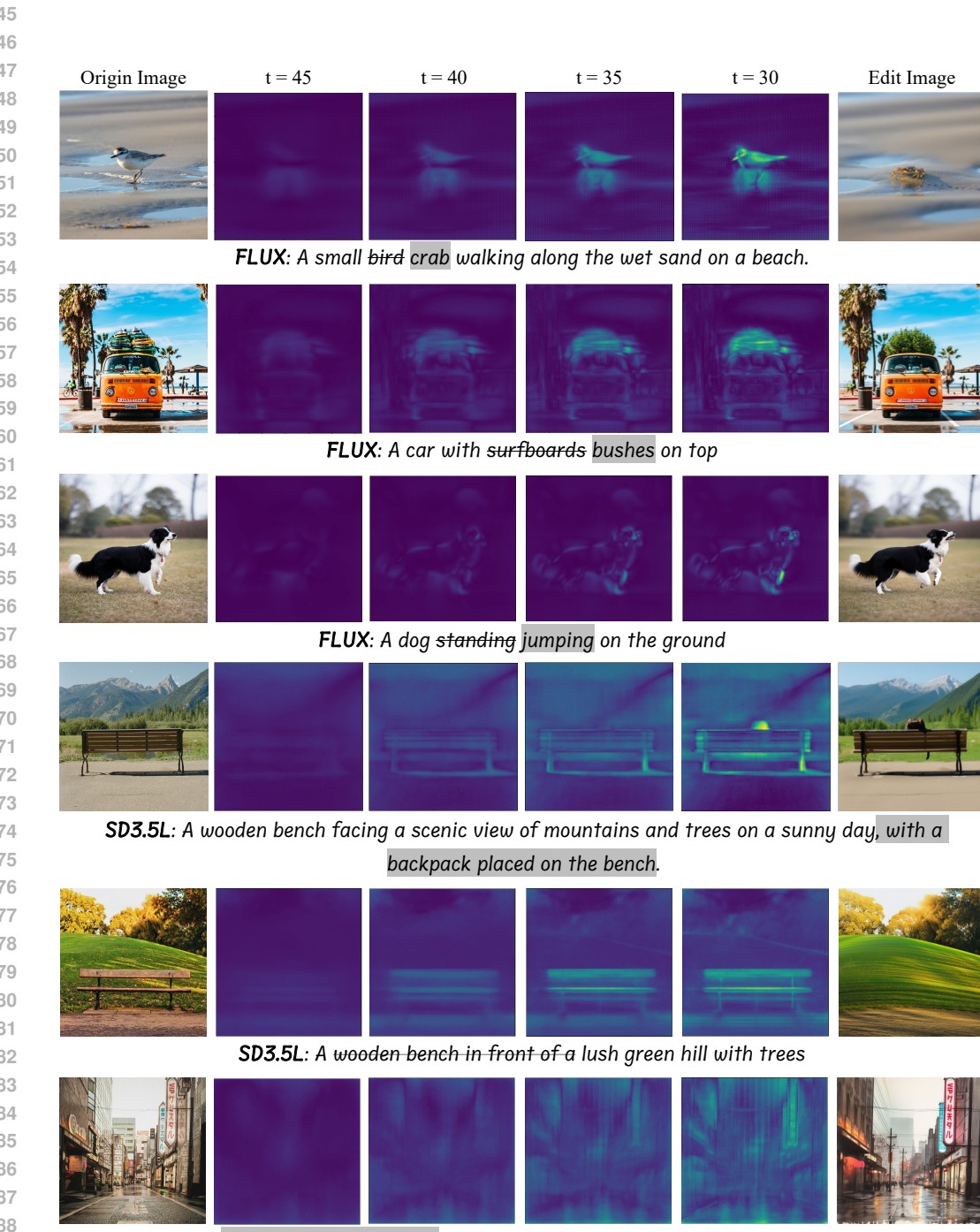

Figure 16: **Visualization of bypass under different** $t_B$**.** Zoom in for a better view.

