# OpenReview forum: "FlowBypass: General Training-Free Rectified Flow Image Editing via Trajectory Bypass"
_ICLR.cc/2026/Conference — Submitted to ICLR 2026_

### Official Review · Reviewer_bETU · 2025-10-27

**Soundness:** 2
**Presentation:** 2
**Contribution:** 2
**Rating:** 4
**Confidence:** 5

**Summary:**

This paper introduces FlowBypass, a novel, general, and training-free image editing framework built upon the Rectified Flow (RF) model. The core motivation is to resolve the inherent trade-off in existing inversion-reconstruction editing pipelines: long trajectories preserve prompt alignment but accumulate discretization errors, compromising fidelity; short trajectories maintain fidelity but lack editing alignment.

FlowBypass addresses this by analytically deriving a "trajectory bypass" ($b_t$) as the solution to a first-order linear Ordinary Differential Equation (ODE). This bypass directly connects an intermediate state of the inversion trajectory ($x_{t_B}$) to the reconstruction trajectory ($y_{t_B} = x_{t_B} + b_{t_B}$), circumventing the error-prone terminal noise state. The framework achieves editing without relying on backbone-specific feature manipulation, which enhances its generality. The authors employ Euler discretization and practical approximations (e.g., finite-difference approximation for the partial derivative of the velocity field) to implement the analytical solution.

Extensive experiments on the EditEvalv2 benchmark, utilizing various RF backbones (SD3.5M, SD3.5L, FLUX.1-dev), demonstrate that FlowBypass consistently achieves a superior balance between prompt alignment and image fidelity compared to state-of-the-art methods.

**Strengths:**

Strong and Novel Theoretical Foundation: The central contribution, the "trajectory bypass" ($b_t$), is rigorously derived as the analytical solution to a linear ODE (Equation 10). This principled approach provides a solid, non-heuristic framework for error mitigation in inversion-based editing, which is a significant theoretical advance over prior empirical or feature-manipulation-based methods.

Excellent Generalizability: By abstaining from backbone-specific Feature Manipulation (FM), the method successfully demonstrates robust performance across diverse Rectified Flow architectures (SD3.5M/L and FLUX.1-dev). This general applicability makes FlowBypass a potentially influential technique in the broader field of diffusion model editing.

Competitive State-of-the-Art Performance: The quantitative results clearly position FlowBypass favorably on the fidelity-alignment Pareto front (Figure 5). It achieves superior perceptual fidelity (lowest LPIPS on FLUX.1-dev) and strong text alignment (best T.Sim. on SD3.5L), confirming its effectiveness in achieving the intended balance.

Thorough Ablation Studies: The authors provide comprehensive ablation studies on critical design choices, including the impact of the bypass step $t_B$ (Figure 10, Table 6), the choice of prompt conditionings (Figure 6, Table 4), and the necessity of the analytical approximations (Table 3, Figure 8). These experiments solidify the robustness and rationale behind the proposed framework's final configuration.

**Weaknesses:**

High Computational Overhead due to Gradient Approximation: The core mechanism relies on approximating the partial derivative of the velocity field, $\frac{\partial}{\partial x_{t}}\hat{v}_{\theta}$, using finite differences (Equation 12). This typically requires two forward passes of the entire generative model per step to calculate the derivative, which can be computationally prohibitive, especially for large models like SD3.5L and FLUX.1-dev. The paper lacks a detailed analysis or comparison of the runtime and computational cost overhead introduced by this specific approximation compared to standard inference methods.

Impact of Simplistic Approximations: While the approximations (especially the first-order Taylor expansion for the exponential term in Equation 13) are crucial for numerical stability, relying on such simplistic methods can potentially limit the accuracy of the bypass calculation, particularly in highly non-linear regions of the velocity field. The trade-off between stability/efficiency and precision should be discussed more critically.

Sensitivity to Hyperparameters ($t_B$ and CFG Scale): The ablation studies (Figure 9, Figure 10) show that performance is highly sensitive to the choice of the bypass timestep ($t_B$) and the CFG scale ($\omega$). While $t_B=30$ and $\omega=2.0$ are selected as optimal, the need for extensive search over these parameters for optimal performance suggests that FlowBypass may require careful tuning for new models or editing tasks. A more adaptive or automated way to select these parameters would significantly improve the practical utility of the method.

**Questions:**

Above

---

> ### Author Response · Authors · 2025-11-21
> **Response bETU (Part 1)**
>
> ## W1. High Computational Overhead due to Gradient Approximation
> The reviewer’s observation regarding computational overhead is correct. The calculation of bypass corresponds to an overhead of approximately **21.1%** compared with the standard inversion–reconstruction pipeline theoretically, primarily due to the additional inference steps needed to compute $b_{t_B}$.
>
> In the revision, we included a runtime comparison against several SOTA baselines, as shown in Table W1.1 (**Table 7 in the revision**).
> Besides, we break down the computational cost into various stages, as shown in Table W1.2 (**Table 8 in the revision**). The discrepancy between the totals in Table W1.1 and Table W1.2 arises because the text encoding time is excluded in Table W1.2.
>
>
> **Table W1.1** Runtime comparison with SOTA editing methods at 1024 $\times$ 1024 resolution. The \# in parentheses indicates the ranking.
> | Method | Backbone | Precision | 4090 Runtime (s)↓ | L20 Runtime (s)↓ |
> | ------- | ------- | ------- | ------- | ------- |
> | P2P* | SD1.4 | FP32 | 17.59 (\#3) | 24.41 (\#4) |
> | NTI | SD1.4 | FP32 | 424.03 (\#14) | 651.65 (\#14) |
> | DDCM | LCM v7 | FP16 | 5.36 (\#1) | 6.47 (\#1) |
> | IP2P | SD1.4 | FP32 | 8.28 (\#2) | 22.16 (\#3) |
> | Omni-Gen | Phi-3 | FP32 | 81.59 (\#8) | 101.84 (\#11) |
> | LEDITS++ | SD1.5 | FP32 | 33.19 (\#6) | 45.18 (\#7) |
> | RF-Solver | FLUX.1-dev | BF16 | 256.23 (\#13) | 138.33 (\#12) |
> | RF-Inversion | FLUX.1-dev | BF16 | 117.24 (\#10) | 51.78 (\#8) |
> | FluxSpace* | FLUX.1-dev | BF16 | 145.50 (\#11) | 65.75 (\#9) |
> | FireFlow | FLUX.1-dev | BF16 | 29.85 (\#5) | 20.66 (\#2) |
> | FlowEdit | FLUX.1-dev | BF16 | 101.08 (\#9) | 43.95 (\#6) |
> | **FlowBypass** | SD3.5 Medium | FP16 | 29.43 (\#4) | 35.90 (\#5) |
> | **FlowBypass** | SD3.5 Large | FP16 | 64.11 (\#7) | 81.31 (\#10) |
> | **FlowBypass** | FLUX.1-dev | BF16 | 172.24 (\#12) | 192.23 (\#13) |
>
> **Table W1.2** Computational cost breakdown (second). The % in parentheses indicates the percentage.
> | Stage | SD35M | SD35L | FLUX.1 |
> | ------- | ------- | ------- | ------- |
> | VAE encode | 0.23 (1.08%) | 0.16 (0.31%) | 0.22 (0.13%) |
> | Inversion | 9.71 (46.25%) | 24.38 (46.97%) | 78.50 (46.52%) |
> | **Inversion for Bypass** | 4.32 (20.56%) | 10.84 (20.87%) | 34.89 (20.67%) |
> | **Bypass** | 0.00836 (0.0398%) | 0.00335 (0.006461%) | 0.00359 (0.002129%) |
> | Recon | 6.42 (30.56%) | 16.27 (31.34%) | 54.90 (32.53%) |
> | VAE decode | 0.32 (1.51%) | 0.26 (0.50%) | 0.24 (0.14%) |
>
>
> Although computing bypass $b_t$ introduces some additional overhead, which means FlowBypass is not the fastest method, the overall runtime remains well within the acceptable range for single-image high-resolution editing.
>
>
> ## W2. Impact of Simplistic Approximations
> FlowBypass involves three simplistic approximations.
>
> **The first approximation** is the linear approximation in Equation 8, which is unavoidable because the actual computation of $y_t$ accumulates errors and would lead to low-fidelity editing, and also motivates the design of the bypass.
>
> **The second approximations** is the derivative approximation in Equation 12, introduced to reduce the memory overhead of gradient computation in large networks. To assess the stability of this approximation, we have added a new ablation study on the derivative factor $\zeta$, as shown in Table W2.1 (**Table 9 in the revision**). The results indicate that varying $\zeta$ has only a minor impact on editing performance, demonstrating the robustness of this approximation.
>
> **Table W2.1** Impact of $\zeta$. [F] indicates the result from FLUX.1-dev, [S] indicates the result from SD3.5 Large.
> | $\zeta$ | [F] LPIPS↓ | [F] I.Sim.↑ | [F] T.Sim.↑ | [S] LPIPS↓ | [S] I.Sim.↑ | [S] T.Sim.↑ |
> | ------- | ------- | ------- | ------- | ------- | ------- | ------- |
> | 0.001 | 0.3290 | 88.56 | 25.66 | 0.4513 | 84.77 | 27.10 |
> | 0.005 | 0.3446 | 87.91 | 25.77 | 0.4488 | 85.09 | 27.14 |
> | **0.01** | 0.3425 | 88.06 | 25.65 | 0.4507 | 84.73 | 27.09 |
> | 0.05 | 0.3509 | 88.03 | 25.84 | 0.4606 | 84.65 | 27.21 |
> | 0.1 | 0.3434 | 88.25 | 25.82 | 0.4562 | 84.81 | 27.32 |
>
> **The third approximation** is the first-order Taylor expansion applied to the positive exponential term in Equation 13. This approximation is primarily introduced to ensure numerical stability, as demonstrated in Table 3 and Figure 8. It is also practically unavoidable, since removing this approximation leads to severe artifacts.
>
> Importantly, owing to the inherent robustness of the denoiser network, sampling along the reconstruction trajectory after applying the bypass can partially compensate for the inaccuracies introduced by these approximations, thereby preserving both fidelity and alignment in the edited results.
>
> In the revision, we have added discussion on the role of these approximations and the trade-offs they entail between stability and precision.

---

> ### Author Response · Authors · 2025-11-21
> **Response bETU (Part 2)**
>
> ## W3. Sensitivity to hyperparameters
> Our current choice of $t_B = 30$ and CFG scale $\omega = 2.0$ has been empirically validated as a robust combination, performing well across most standard image editing tasks. Notably, the experiments reported in Table 1 for three different backbones all use this same parameter setting, demonstrating the stable performance of such combination.
>
> We commit that the selection of $t_B$ and CFG scale $\omega$ is currently heuristic, and developing an adaptive or automated parameter selection mechanism represents a meaningful direction for future work, which could further enhance FlowBypass’s practical applicability to new models or editing tasks.
>
> A practical plug-and-play strategy for adaptive hyperparameter selection is to adjust $t_B$ and the CFG scale $\omega$ based on either user feedback or objective evaluation signals. In general, a smaller $t_B$ tends to emphasize fidelity to the origin image, whereas a larger CFG scale $\omega$ more strongly enforces alignment with the edit prompt. In real use cases, these two parameters can be selectively tuned in response to user-specific preferences.
> For example, increasing $\omega$ when the user desires stronger edits, or decreasing $t_B$ when preserving fine structures is more important.
> We also acknowledge that such a search process may incur non-trivial computational cost, and developing a more automated or efficient adaptation mechanism remains an interesting direction for future work.

---

### Official Review · Reviewer_z99e · 2025-10-27

**Soundness:** 2
**Presentation:** 3
**Contribution:** 2
**Rating:** 4
**Confidence:** 4

**Summary:**

This paper introduces FlowBypass, a training-free image editing framework leveraging Rectified Flow models. By analytically constructing a “bypass” trajectory between inversion and reconstruction, FlowBypass aims to retain high-fidelity details in unedited regions while improving prompt alignment. Extensive quantitative and qualitative experiments demonstrate the superior performance compared to existing training-free methods.

**Strengths:**

- The motivation of this paper is clear and meaningful: Most inversion-based image editing methods suffer from extensive hyper-parameter tuning to strike a balance between image fidelity and prompt alignment, which is a long-lasting problem for training-free image editing methods.
- The proposed method is not tied to specific backbone architectures, which can be applied on various base models such as FLUX and SD3.
- Quantitative and qualitative results in paper demonstrate that the outstanding performance of the proposed method.
- The paper is clear and well-written.

**Weaknesses:**

- Missing several baselines: FlowEdit [1], QK-Edit [2], FluxSpace [3], FireFlow [4].
- Intuitively, given a source image, one could invert it to a specific time step $t$ and then start denoising from that step to obtain the edited image. This may also achieve high-fidelity image editing. However, in Table 2, authors only provide the results of $t=30$ and $t=50$, which is not enough to illustrate the advantages of the proposed method. Authors should provide more comprehensive comparisons between their method and this approach in ablation study (such as $t=40$).
- As illustrated in Line 256, FlowBypass constructs the reconstruction trajectory starting from an intermediate state rather than from the inverted noise. Can this make FlowBypass more efficient than previous methods? Authors are expected to provide a comparison of runtimes.


[1]. FlowEdit: Inversion-Free Text-Based Editing Using Pre-Trained Flow Models

[2]. QK-Edit: Revisiting Attention-based Injection in MM-DiT for Image and Video Editing

[3]. FluxSpace: Disentangled Semantic Editing in Rectified Flow Transformers

[4]. FireFlow: Fast Inversion of Rectified Flow for Image Semantic Editing

**Questions:**

See weakness.

---

> ### Author Response · Authors · 2025-11-21
> **Response z99e (Part 1)**
>
> ## W1. More compared baselines
> We added four new baselines in the **Table 1 of the revision**, including **FlowEdit**, **FluxSpace**, and **FireFlow**. Among them, FluxSpace natively supports only synthetic-image editing; real-image editing requires obtaining the noise via RF-Inversion. Since the authors have not released code for this setting, we implemented the corresponding pipeline ourselves and included it in the comparison. QK-Edit has not released its source code, so it could not be incorporated into our experiments.
>
> Across all newly added baselines, FlowBypass consistently delivers superior performance both in overall editing fidelity and in prompt alignment, and continues to remain on the Pareto frontier in the fidelity-alignment trade-off, as demonstrated in Table W1.1.
>
> **Table W1.1** Comparison with state-of-the-art image editing methods. The * marks indicate that the official code lacks real-image editing implementations, so we implement them ourselves following the authors’ guidance.
> | Method | Backbone | LPIPS↓ | I.Sim.↑ | T.Sim.↑ |
> | ------- | ------- | ------- | ------- | ------- |
> | P2P* (New) | SD1.4 | 0.4990 | 81.04 | 26.93 |
> | NTI | SD1.4 | 0.5798 | 73.96 | 26.38 |
> | DDCM | LCM v7 | 0.4507 | 87.14 | 26.62 |
> | IP2P | SD1.4 | 0.6103 | 84.85 | 23.95 |
> | Omni-Gen | Phi-3 | 0.3573 | 87.48 | 25.58 |
> | LEDITS++ | SD1.5 | 0.3554 | 81.54 | 21.73 |
> | RF-Solver | FLUX.1-dev | 0.3880 | 87.32 | 25.30 |
> | RF-Inversion | FLUX.1-dev | 0.5659 | 83.35 | 25.71 |
> | FluxSpace* (New) | FLUX.1-dev | 0.8058 | 79.74 | 22.74 |
> | FireFlow (New) | FLUX.1-dev | 0.3850 | 87.01 | 25.69 |
> | FlowEdit (New) | FLUX.1-dev | 0.3921 | 87.90 | 25.28 |
> | **FlowBypass** | SD3.5 Medium | 0.4228 | 85.96 | 26.45 |
> | **FlowBypass** | SD3.5 Large | 0.4507 | 84.73 | **27.09** |
> | **FlowBypass** | FLUX.1-dev | **0.3425** | **88.06** | 25.65 |
>
>
> ## W2. Ablation on reconstruction without bypass
> We thank the reviewer for the suggestion. Directly starting reconstruction from a fixed timestep $t$ can easily lead to editing failures, as it essentially begins reconstruction on an incorrect trajectory. In the initial manuscript, we only reported results for $t=30$ (corresponding to the default $t_B$) and $t=50$ (the total number of steps) as illustrative examples.
>
> In the revision, we include an expanded table showing results for $t=10, 20, 30, 40, 50$. When starting from larger $t$, fidelity is difficult to maintain, whereas starting from smaller $t$ compromises alignment, as illustrated in Table W2.1 (**Table 10 in the revision**). **FlowBypass addresses this by correctly jumping the reconstruction starting point onto the reconstruction trajectory, achieving a balance between fidelity and alignment and yielding more stable, high-quality edits.**
>
> **Table W2.1** Impact of reconstruction from different $t$ without bypass.  [F] indicates the result from FLUX.1-dev, [S] indicates the result from SD3.5 Large.
> | Rec. t | [F] LPIPS↓ | [F] I.Sim.↑ | [F] T.Sim.↑ | [S] LPIPS↓ | [S] I.Sim.↑ | [S] T.Sim.↑ |
> | ------- | ------- | ------- | ------- | ------- | ------- | ------- |
> | 10 | 0.1585 | 97.56 | 22.35 | 0.2561 | 95.50 | 22.81 |
> | 20 | 0.1834 | 96.94 | 22.88 | 0.2817 | 94.31 | 23.82 |
> | 30 | 0.2240 | 94.47 | 23.85 | 0.3288 | 91.01 | 25.36 |
> | 40 | 0.3358 | 88.32 | 25.53 | 0.4487 | 84.43 | 26.72 |
> | 50 | 0.5811 | 78.21 | 27.78 | 0.6576 | 76.95 | 27.94 |
> | **30 w/ Bypass** |  0.3425 | 88.06 | 25.65 | 0.4507 | 84.73 | 27.09 |

---

> ### Author Response · Authors · 2025-11-21
> **Response z99e (Part 2)**
>
> ## W3. Runtime comparison
> The calculation of bypass corresponds to an overhead of approximately **21.1%** compared with the standard inversion–reconstruction pipeline theoretically, primarily due to the additional inference steps needed to compute $b_{t_B}$.
>
> In the revised version, we included a runtime comparison against several SOTA baselines, as shown in Table W3.1 (**Table 7 in the revision**).
> We conduct experiments in the manuscript on an RTX 4090. For FLUX-based baselines, we employ DeepSpeed to overcome VRAM limitations. To ensure fairness, we report runtimes on both an RTX 4090 with DeepSpeed and an L20 without DeepSpeed.
>
>
> **Table W3.1** Runtime comparison with SOTA editing methods at 1024 $\times$ 1024 resolution. The \# in parentheses indicates the ranking.
> | Method | Backbone | Precision | 4090 Runtime (s)↓ | L20 Runtime (s)↓ |
> | ------- | ------- | ------- | ------- | ------- |
> | P2P* | SD1.4 | FP32 | 17.59 (\#3) | 24.41 (\#4) |
> | NTI | SD1.4 | FP32 | 424.03 (\#14) | 651.65 (\#14) |
> | DDCM | LCM v7 | FP16 | 5.36 (\#1) | 6.47 (\#1) |
> | IP2P | SD1.4 | FP32 | 8.28 (\#2) | 22.16 (\#3) |
> | Omni-Gen | Phi-3 | FP32 | 81.59 (\#8) | 101.84 (\#11) |
> | LEDITS++ | SD1.5 | FP32 | 33.19 (\#6) | 45.18 (\#7) |
> | RF-Solver | FLUX.1-dev | BF16 | 256.23 (\#13) | 138.33 (\#12) |
> | RF-Inversion | FLUX.1-dev | BF16 | 117.24 (\#10) | 51.78 (\#8) |
> | FluxSpace* | FLUX.1-dev | BF16 | 145.50 (\#11) | 65.75 (\#9) |
> | FireFlow | FLUX.1-dev | BF16 | 29.85 (\#5) | 20.66 (\#2) |
> | FlowEdit | FLUX.1-dev | BF16 | 101.08 (\#9) | 43.95 (\#6) |
> | **FlowBypass** | SD3.5 Medium | FP16 | 29.43 (\#4) | 35.90 (\#5) |
> | **FlowBypass** | SD3.5 Large | FP16 | 64.11 (\#7) | 81.31 (\#10) |
> | **FlowBypass** | FLUX.1-dev | BF16 | 172.24 (\#12) | 192.23 (\#13) |
>
> Although computing bypass $b_t$ introduces some additional overhead, which means FlowBypass is not the fastest method, the overall runtime remains well within the acceptable range for single-image high-resolution editing.

---

### Official Review · Reviewer_K56Z · 2025-10-30

**Soundness:** 3
**Presentation:** 3
**Contribution:** 3
**Rating:** 6
**Confidence:** 5

**Summary:**

This paper presents FlowBypass, a training-free image editing framework based on Rectified Flow. The method introduces a "trajectory bypass" to mitigate error accumulation in inversion-reconstruction pipelines, striking a balance between semantic alignment and fidelity preservation in irrelevant regions.

**Strengths:**

The core idea of constructing a bypass between inversion and reconstruction trajectories is well-motivated. The theoretical derivation of the bypass term from first principles is rigorous and elegant. The method effectively addresses the fidelity-alignment trade-off without relying on model-specific feature manipulations, ensuring broader applicability.

**Weaknesses:**

The paper emphasizes superior performance in "challenging editing scenarios" but relies solely on the EditEvalv2 benchmark (150 images). While the appendix includes external examples, it lacks systematic evaluation across diverse domains. Experiments neglect complex edits like multi-object coordination or motion changes. Testing on dynamic edits (e.g., "walking" to "dancing") would better validate temporal consistency and generalization. This narrow validation weakens claims of robustness, particularly for complex edits like structural transformations.

No dedicated section discusses failure cases or boundaries of the method. For instance, Figure 8 reveals artifacts from approximation failures, but the paper does not analyze root causes. Explicitly outlining scenarios where FlowBypass fails would provide a more balanced perspective.

The reliance on automated metrics overlooks human perceptual factors critical for image editing. A user study assessing edit naturalness and consistency would complement quantitative results and align with real-world application needs.

The paper omits runtime comparisons with state-of-the-art methods.

The paper highlights independence from feature manipulations but does not rigorously compare with methods like HeadRouter or Prompt-to-Prompt. Such comparisons are essential to demonstrate advantages in localized editing precision (e.g., texture replacement) and scenario applicability.

**Questions:**

See weaknesses for details.

**Details Of Ethics Concerns:**

The ethics statement mentions adding meta-information to edited images. What specific techniques (e.g., watermarking) are used? How robust are they against removal?

---

> ### Author Response · Authors · 2025-11-21
> **Response K56Z (Part 1)**
>
> ## W1. Performance on action-change sub-task
> Thanks for reviewer's careful review and meaningful suggestions.
> EditEvalv2 is a well-designed dataset for image editing, which includes seven categories of editing sub-tasks, among which the **action-change** sub-task mentioned by the reviewer is explicitly covered. This sub-task involves substantial structural deformation and semantic transformation, making it a key dimension for evaluating complex editing behaviors.
>
> It is important to clarify that FlowBypass is an image editing method and does not operate over the temporal dimension of videos; therefore, notions such as “dynamic edits’’ or “temporal consistency’’ fall outside the scope of our evaluation. Nevertheless, to more fully demonstrate the model’s generalization ability on action-related edits, we have added additional action-change examples in **Figure 13 of the revision**, including *walking → running* and *walking → dancing*. These cases consistently show that FlowBypass handles pose changes and structural adjustments in a stable and reliable manner.
>
> Overall, the task design in EditEvalv2 already covers the type of complex semantic transformations raised by the reviewer, and the newly added action-editing examples further validate the robustness of our method for structural modifications.
>
>
>
> ## W2. Discussion on limitations and failures
> Thanks for the suggestion.
>
> **In terms of limitations**, FlowBypass introduces additional computation due to the bypass term, which results in a noticeable increase in inference cost. Although the method is not the most efficient among image editing approaches, the overall runtime remains acceptable for single-image high-resolution editing. Another limitation concerns negation prompts: when the edit prompt contains expressions such as *“without”*, the instability mainly originates from the backbone generative models (FLUX / SD3.5), which are known to struggle with negative conditioning. This issue is shared by many editing methods built upon these backbones, as the models themselves provide weak and unreliable outputs for negation. FlowBypass inherits this behavior and may occasionally fail in such cases. In practice, edits would become more stable and accurate when the negation is reformulated into direct phrasing without negation, such as transforming *“a cat with a hat”* into simply *“a cat”*.
>
> **As for the failure cases in Figure 8**, we clarify that this visualization corresponds to an ablation study that evaluates the effectiveness of our approximation, instead of a failure case presentation of FlowBypass. The observed failures stem from inaccurate approximations, particularly the approximation of the exponential term in Equ. 13. We argue that these artifacts mainly arise from exponential explosion, which injects excessively large values into the calculated bypass, causing the normalization layers in the denoiser network to behave improperly, and finally introduces these unpleasant artifacts.
>
> According to the reviewer's suggestion, we have added a dedicated Limitations section (**Sec. 5 in the revision**) to systematically discuss the constraints of FlowBypass, and we have expanded the explanation accompanying Figure 8 to clarify the underlying causes of the observed failure cases in **Sec. A.2.2**.
>
> ## W3. Subjective user study
> Thanks for the insight suggestion.
> We additionally conducted an user study to evaluate the naturalness and consistency of the edited results in **Figure 12 of the revision**. As shown in the Table W3.1, most participants preferred the outputs edited by FlowBypass rather than other edit methods. This provides further evidence of FlowBypass’s perceptual advantages from a user perspective and complements automated metrics by covering aspects of subjective visual quality that quantitative evaluations may miss.
>
> **Table W3.1** User study. The * marks indicate that the official code lacks real-image editing implementations, so we implement them ourselves following the authors’ guidance.
> | Method | SD3.5M Prefer | SD3.5L Prefer | FLUX Prefer |
> | ------- | ------- | ------- | ------- |
> | P2P* | 95.45% | 91.67% | 89.19% |
> | NTI | 89.61% | 85.45% | 89.39% |
> | DDCM | 79.59% | 80.82% | 79.17% |
> | IP2P | 84.38% | 81.54% | 86.57% |
> | Omni-Gen | 70.42% | 64.71% | 73.91% |
> | LEDITS++ | 90.16% | 92.00% | 95.16% |
> | RF-Solver | 58.93% | 67.31% | 72.22% |
> | RF-Inversion | 81.03% | 87.88% | 90.00% |
> | FluxSpace* | 93.94% | 96.43% | 93.55% |
> | FireFlow | 77.14% | 68.00% | 90.00% |
> | FlowEdit | 81.58% | 72.41% | 72.73% |

---

> ### Author Response · Authors · 2025-11-21
> **Response K56Z (Part 2)**
>
> ## W4. Runtime comparison
> In the revised version, we included a runtime comparison against several SOTA baselines, as shown in Table W4.1 (**Table 7 in the revision**).
> We conduct experiments in the manuscript on an RTX 4090. For FLUX-based baselines, we employ DeepSpeed to overcome VRAM limitations. To ensure fairness, we report runtimes on both an RTX 4090 with DeepSpeed and an L20 without DeepSpeed.
> Although computing bypass $b_t$ introduces some additional overhead (~21.1% theoretically), which means FlowBypass is not the fastest method, the overall runtime remains well within the acceptable range for single-image high-resolution editing.
>
> **Table W4.1** Runtime comparison with SOTA editing methods at 1024 $\times$ 1024 resolution. The \# in parentheses indicates the ranking.
> | Method | Backbone | Precision | 4090 Runtime (s)↓ | L20 Runtime (s)↓ |
> | ------- | ------- | ------- | ------- | ------- |
> | P2P* | SD1.4 | FP32 | 17.59 (\#3) | 24.41 (\#4) |
> | NTI | SD1.4 | FP32 | 424.03 (\#14) | 651.65 (\#14) |
> | DDCM | LCM v7 | FP16 | 5.36 (\#1) | 6.47 (\#1) |
> | IP2P | SD1.4 | FP32 | 8.28 (\#2) | 22.16 (\#3) |
> | Omni-Gen | Phi-3 | FP32 | 81.59 (\#8) | 101.84 (\#11) |
> | LEDITS++ | SD1.5 | FP32 | 33.19 (\#6) | 45.18 (\#7) |
> | RF-Solver | FLUX.1-dev | BF16 | 256.23 (\#13) | 138.33 (\#12) |
> | RF-Inversion | FLUX.1-dev | BF16 | 117.24 (\#10) | 51.78 (\#8) |
> | FluxSpace* | FLUX.1-dev | BF16 | 145.50 (\#11) | 65.75 (\#9) |
> | FireFlow | FLUX.1-dev | BF16 | 29.85 (\#5) | 20.66 (\#2) |
> | FlowEdit | FLUX.1-dev | BF16 | 101.08 (\#9) | 43.95 (\#6) |
> | **FlowBypass** | SD3.5 Medium | FP16 | 29.43 (\#4) | 35.90 (\#5) |
> | **FlowBypass** | SD3.5 Large | FP16 | 64.11 (\#7) | 81.31 (\#10) |
> | **FlowBypass** | FLUX.1-dev | BF16 | 172.24 (\#12) | 192.23 (\#13) |
>
>
>
> ## W5. More compared baselines and texture modification results
> In the **Table 1 of the revision**, we added four new baselines, including **Prompt-to-Prompt**. It is important to clarify that Prompt-to-Prompt natively supports only synthetic image editing; real-image editing using Prompt-to-Prompt requires DDIM-Inversion, and the authors have noted in their GitHub issues that its robustness is limited, with no official implementation available. We reproduced the real-image editing pipeline following their described procedure and included it in our comparisons. In contrast, **HeadRouter** has no released code, making it infeasible to include as a baseline.
> The comparison is presented in Table W5.1, which demonstrates that FlowBypass still consistently surpasses other methods, and holds the Pareto frontier.
>
> **Table W5.1** Comparison with state-of-the-art image editing methods. The * marks indicate that the official code lacks real-image editing implementations, so we implement them ourselves following the authors’ guidance.
> | Method | Backbone | LPIPS↓ | I.Sim.↑ | T.Sim.↑ |
> | ------- | ------- | ------- | ------- | ------- |
> | P2P* (New) | SD1.4 | 0.4990 | 81.04 | 26.93 |
> | NTI | SD1.4 | 0.5798 | 73.96 | 26.38 |
> | DDCM | LCM v7 | 0.4507 | 87.14 | 26.62 |
> | IP2P | SD1.4 | 0.6103 | 84.85 | 23.95 |
> | Omni-Gen | Phi-3 | 0.3573 | 87.48 | 25.58 |
> | LEDITS++ | SD1.5 | 0.3554 | 81.54 | 21.73 |
> | RF-Solver | FLUX.1-dev | 0.3880 | 87.32 | 25.30 |
> | RF-Inversion | FLUX.1-dev | 0.5659 | 83.35 | 25.71 |
> | FluxSpace* (New) | FLUX.1-dev | 0.8058 | 79.74 | 22.74 |
> | FireFlow (New) | FLUX.1-dev | 0.3850 | 87.01 | 25.69 |
> | FlowEdit (New) | FLUX.1-dev | 0.3921 | 87.90 | 25.28 |
> | **FlowBypass** | SD3.5 Medium | 0.4228 | 85.96 | 26.45 |
> | **FlowBypass** | SD3.5 Large | 0.4507 | 84.73 | **27.09** |
> | **FlowBypass** | FLUX.1-dev | **0.3425** | **88.06** | 25.65 |
>
>
> Regarding texture replacement, EditEvalv2 already contains a **texture change** sub-task. We further supplemented the revision with additional localized texture replacement examples in **Figure 13 of the revision**, providing a more comprehensive evaluation of FlowBypass in terms of localized editing precision and applicability across diverse scenarios.
>
> ## Details Of Ethics Concerns
> In our edited outputs, we apply an invisible watermark and also record edit information in the image file metadata, marking the image as AI-edited, like other edit approaches. These indicators can be generally preserved under typical image transmission pipelines and routine editing operations, though we acknowledge they are not designed to withstand deliberate removal.
>
> Our aim is not to propose a tamper-resistant watermarking scheme, which is outside the scope of our work, but to ensure that edited images carry clear and identifiable indicators of modification under normal usage conditions. We have explicitly stated these design choices.

---

### Official Review · Reviewer_7EG6 · 2025-10-31

**Soundness:** 2
**Presentation:** 3
**Contribution:** 2
**Rating:** 4
**Confidence:** 5

**Summary:**

The authors present the idea that the standard inversion process may not identify the optimal reversed noise for subsequent editing tasks. Instead, they propose initiating the reconstruction process from an alternative starting point, determined by computing an offset that bridges the inversion and image generation processes. This offset is formulated as a linear differential equation with an analytical solution. Through appropriate approximation, they demonstrate that this new starting point contributes to preserving high-fidelity details during prompt-guided image editing. Results on benchmark datasets appear to support the effectiveness of the proposed method.

**Strengths:**

- The training-free image editing framework, that leverages a trajectory bypass mechanism to achieve superior fidelity-alignment trade-offs, seems show strong generalizability across Rectified Flow models.
- The core idea appears theoretically sound, and the experimental results demonstrate promising performance.

**Weaknesses:**

1. The derivation from Equation (5) to Equation (6) appears to suggest that $x_1 = y_1$. However, since $y_t$ corresponds to the reconstruction process and $x_t$ corresponds to the inversion process, it seems more natural that $x_1$ refers to the original image while $y_1$ represents the reversed noise. If the authors intend for both $x_1$ and $y_1$ to represent the original image, could they clarify why the integration bounds and velocity direction (should it be $-\hat{v}_\theta$?) are set as shown? Please correct me if I have misunderstood this derivation.

2. The formulation of FlowBypass from Equation (5) to Equation (7) appears very similar to FlowEdit [1] when differentiating Equation (5). Specifically, the key replacement in Equation (7), where $y_t = x_t + b_t$, corresponds to Equation (7) in [1]. While the authors clearly draw inspiration from related work, FlowEdit is not cited in the manuscript. Additionally, while the introduction of $dZ_t^{inv}$ (equivalent to $\frac{d}{dt}b_t$) in [1] is quite intuitive, the integration-based derivation in this paper makes the subtraction of $y_t$ and $x_t$ less clear. Could the authors clarify the relationship to FlowEdit and explain the conceptual advantage of their formulation?

3. The derivation of the bypass term (Equation 8) assumes that the offset $b_t$ is small enough to permit first-order Taylor expansion. This assumption may not hold for significant edits (e.g., drastic content changes), potentially affecting the accuracy of the bypass calculation. The paper would benefit from discussing when this assumption is valid and how the method performs when it breaks down.

4. Given the similarity to FlowEdit [1], it should be included as a baseline for comparison. Additionally, considering the extra computational cost of calculating $b_t$, comparing with efficient methods such as [2] would provide a more complete picture of the speed-performance trade-off.

5. The paper would benefit from a more thorough discussion of the method's limitations, particularly its performance in challenging scenarios such as extreme content modifications or style transfer tasks.

6. The authors claim that prompt choice has a crucial impact on the final results. This raises the question of whether baseline methods could achieve better performance if the same prompt optimization strategy were applied to them, since prompt selection is independent of the method itself. Without this ablation study on baseline methods, the comparison may not be entirely fair.

## Reference:
- [1] Flowedit: Inversion-free text-based editing using pre-trained flow models. ICCV2025
- [2] Fireflow: Fast inversion of rectified flow for image semantic editing. ICML2025

**Questions:**

- Ablation Study on Prompt Configuration (Lines 228-230): The use of $C^p_{rec}=C_y$ and $C^n_{rec}=C_x$ in the reconstruction setting appears to be an important design choice. Could the authors conduct an ablation study to evaluate this configuration? Specifically, what happens if the negative prompt is left as null text instead?

- The paper lacks ablation studies on two key design choices in Equation (13): (1) the selection of the hyperparameter $\xi$, and (2) the decision to replace the exponential with its first-order Taylor approximation in the positive domain. It would be valuable to understand how sensitive the method is to these choices and what motivated this particular approximation.

-  According to Section 3.4, the key step relies on the construction of $b*_{t_i}$. It would be helpful to include visualizations showing the intermediate results of $b*_{t_i}$ or $y_{t_B}$ to provide better intuition about how the bypass mechanism affects the trajectory.

- How many model forward passes are required for computing $b^*_{t_i}$ and completing the entire image editing process? A detailed breakdown of the computational overhead would help readers assess the practical efficiency of the method, especially compared to baseline approaches.

---

> ### Author Response · Authors · 2025-11-21
> **Response 7EG6 (Part 1)**
>
> ## W1. Notation explaination of $x_t$ and $y_t$
> Thanks for the reviewer’s careful reading and helpful comments. We clarify the notation and the integration direction below to avoid misunderstanding.
>
> **In short**, $x_0$ denotes the original image, $y_0$ denotes the edited image, $x_1 = y_1 = \epsilon$ denotes the shared terminal noise, and $x_t$ and $y_t$ denote the states along the inversion and reconstruction trajectories, respectively.
>
> Here, we explain the notations in detail.
>
> **First**, the subscript $t$ in $x_t$ and $y_t$ simply denotes the ODE timestep along their respective trajectories, following the convention in the rectified flow literature. For the forward noising process of RF, we have
> $$
> x_t = t \cdot \epsilon + (1-t) \cdot x_0.
> $$
> Thus $x_1$ corresponds to pure noise and $x_0$ to the input image. Similarly, $y_0$ denotes the reconstructed (edited) target image. In inversion-based editing pipelines, one typically inverts an input image to a shared noise endpoint before reconstructing from that noise. Therefore, we explicitly set $x_1 = y_1$, as stated in Line 151 of initial manuscript or Line 155 of revision .
>
> **Second**, we explain integration bounds and velocity direction.
> Equ. 5 is derived from Equ. 3 by setting $r=1$.
> Equ. 3 describes the general RF ODE and does not impose any ordering between $t$ and $r$, since ODE flows are reversible. Throughout the paper, $\hat v_{\theta}$ consistently denotes the velocity field directing from noise to image, following standard RF formulations. Based on this, Euler discretization produces the inversion (Equ. 11) and reconstruction (Equ. 14) updates from the monotonically increasing timestep sequence $\{t_i\}$.
> During inversion, the factor $(t_{i+1}-t_i)>0$ pushes the state towards noise, whereas during reconstruction the factor $(t_{i-1}-t_i)<0$ pushes towards the image $y_0$.
> Thus, the direction of the velocity is determined not by any manually assigned sign, but inherently by the sign of the timestep differences in $\{t_i\}$.
>
> We have further clarified the notation and integration directions in the final version to avoid potential confusion. We hope this addresses the reviewer’s concerns.
>
> ## W2. The differences and advantages from FlowEdit
> Thanks for raising this point. We clarify the relationship and differences between FlowBypass and FlowEdit here. Although some expressions may appear similar, the two methods differ fundamentally in motivation, trajectory construction, derivation, and editing mechanism.
>
> ### W2.1. Different method categories
>
> * FlowEdit is an inversion-free approach, which does not perform inversion on the input image. Instead, it starts from the origin image $x_0$ and iteratively adjusts $x_t$ under the RF linearity assumption to achieve editing.
>
> * FlowBypass is a complete inversion-based pipeline, which first invert the input image to a noisy middle state, then reconstruct after conducting bypass on it. The bypass operation happens only once between the inversion and reconstruction trajectories.
>
> Thus, the two methods belong to fundamentally different categories.
>
> ### W2.2. Different meanings of $x_t$ and $y_t$
>
> * In FlowEdit, there is no independent $y_t$ trajectory. $y_0$ is the reconstruction endpoint, while $x_t$ is a theoretical intermediate variable derived from linearity. The term $y_t - x_t$ is only used to construct a corrective update and does not refer to two actual trajectories.
>
> * In FlowBypass, $x_t$ (inversion trajectory) and $y_t$ (reconstruction trajectory) are two real ODE trajectories. We apply a first-order Taylor expansion around $x_t$ and compensate for the residual error to obtain Equ. 7 to mitigate the error accumulation of vanilla $y_t$.
>
> ### W2.3. Conceptual advantages of FlowBypass
>
> Since FlowEdit performs direct jumps between inherently more complex image distributions, its editing results may suffer from undesirable artifacts, as illustrated in the corresponding results in **fourth row of Figure 2, seventh and eighth rows of Figure 11**. In contrast, FlowBypass first maps the input into a noisy representation via inversion and later reconstructs the output, thereby conducting a single, direct jump within a comparatively simpler image–noise hybrid distribution. The subsequent refinement by the generative model further mitigates such artifacts, leading to a significant reduction in editing-induced artifacts.
>
> In summary, despite superficial notational similarities, FlowBypass and FlowEdit differ fundamentally in objectives, mechanisms, trajectory semantics, and theoretical derivations. We hope this explaination could make this distinction clearer and have add the relevant citations in the revision.

---

> ### Author Response · Authors · 2025-11-21
> **Response 7EG6 (Part 2)**
>
> ## W3. Validity limits of first-order bypass approximation
> Thanks for raising this important point. We fully agree that when the bypass $b_t$ becomes large, the first-order Taylor approximation in Equ. 8 may lose accuracy and affect the bypass quality. We clarify this assumption and its practical implications below.
>
> **On the one hand**, the magnitude of $b_t$ is closely tied to the choice of $t_B$. Near the noise end of the trajectory (larger $t_B$), the inversion and reconstruction paths remain close and the bypass stays small; near the image end (smaller $t_B$), the two trajectories diverge, leading to a larger $b_t$. Thus, choosing an excessively small $t_B$ can indeed cause the Taylor approximation to break down.
> As shown in Figure 15, poor approximation accuracy caused by inadequate $t_B$ is clearly reflected in the edited results, leading to insufficient editing.
>
> **On the other hand**, significant edits inherently introduce larger semantic shifts between the inversion and reconstruction trajectories, which naturally increases the bypass $b_t$. In such extreme cases, the deviation cannot be fully compensated by a first-order correction, and the bypass may exhibit noticeable degradation.
>
> **In practice**, most reasonable settings of $t_B$ lie in a wide range, where the two trajectories align well and $b_t$ remains small. In these scenarios, the first-order Taylor expansion is sufficiently accurate, and the bypass remains stable and reliable across a wide variety of typical editing tasks.
> We have conduct an ablation study on hyperparameter $\zeta$ in Equ. 12, as shown in Table Q2.1 in the response of Question 2 (**Table 9 in the revision**).
> The results indicate that varying $\zeta$ has only a minor impact on editing performance, demonstrating the robustness of this approximation, which partially validate the assumption.
>
> When the approximation unfortunately breaks down, a practical remedy is to choose a larger value of $t_B$, which reduces the magnitude of $b_t$ and improves the approximation quality under the risk of sacrificing fidelity because of longer trajectory and larger error accumulation.

---

> ### Author Response · Authors · 2025-11-21
> **Response 7EG6 (Part 3)**
>
> ## W4. More baselines and runtime comparison
> Thanks for the helpful suggestion. In our updated experiments, we have incorporated four additional baselines, including FlowEdit. The extended comparisons show that FlowBypass consistently maintains superior editing quality and positioned in the Pareto frontier, as illustrated in Table W4.1 (**Table 1 in the revision**).
>
> **Table W4.1** Comparison with state-of-the-art image editing methods. The * marks indicate that the official code lacks real-image editing implementations, so we implement them ourselves following the authors’ guidance.
> | Method | Backbone | LPIPS↓ | I.Sim.↑ | T.Sim.↑ |
> | ------- | ------- | ------- | ------- | ------- |
> | P2P* (New) | SD1.4 | 0.4990 | 81.04 | 26.93 |
> | NTI | SD1.4 | 0.5798 | 73.96 | 26.38 |
> | DDCM | LCM v7 | 0.4507 | 87.14 | 26.62 |
> | IP2P | SD1.4 | 0.6103 | 84.85 | 23.95 |
> | Omni-Gen | Phi-3 | 0.3573 | 87.48 | 25.58 |
> | LEDITS++ | SD1.5 | 0.3554 | 81.54 | 21.73 |
> | RF-Solver | FLUX.1-dev | 0.3880 | 87.32 | 25.30 |
> | RF-Inversion | FLUX.1-dev | 0.5659 | 83.35 | 25.71 |
> | FluxSpace* (New) | FLUX.1-dev | 0.8058 | 79.74 | 22.74 |
> | FireFlow (New) | FLUX.1-dev | 0.3850 | 87.01 | 25.69 |
> | FlowEdit (New) | FLUX.1-dev | 0.3921 | 87.90 | 25.28 |
> | **FlowBypass** | SD3.5 Medium | 0.4228 | 85.96 | 26.45 |
> | **FlowBypass** | SD3.5 Large | 0.4507 | 84.73 | **27.09** |
> | **FlowBypass** | FLUX.1-dev | **0.3425** | **88.06** | 25.65 |
>
>
>
>
> Regarding computational efficiency, we agree that the calculation of bypass $b_t$ introduces about 21.1% additional computing burden theoretically, and FlowBypass is therefore not the most efficient training-free solution. As suggested by the reviewers, we provide additional runtime comparisons in Table W4.2 (**Table 7 in the revision**).
> We conduct experiments in the manuscript on an RTX 4090. For FLUX-based baselines, we employ DeepSpeed to overcome VRAM limitations. To ensure fairness, we report runtimes on both an RTX 4090 with DeepSpeed and an L20 without DeepSpeed.
>
> Although FlowBypass is not the fastest method in absolute runtime, it achieves a practical processing cost for high-resolution image editing while offering superior performance and a well balance between fidelity and alignment.
>
> **Table W4.2** Runtime comparison with SOTA editing methods at 1024 $\times$ 1024 resolution. The \# in parentheses indicates the ranking.
> | Method | Backbone | Precision | 4090 Runtime (s)↓ | L20 Runtime (s)↓ |
> | ------- | ------- | ------- | ------- | ------- |
> | P2P* | SD1.4 | FP32 | 17.59 (\#3) | 24.41 (\#4) |
> | NTI | SD1.4 | FP32 | 424.03 (\#14) | 651.65 (\#14) |
> | DDCM | LCM v7 | FP16 | 5.36 (\#1) | 6.47 (\#1) |
> | IP2P | SD1.4 | FP32 | 8.28 (\#2) | 22.16 (\#3) |
> | Omni-Gen | Phi-3 | FP32 | 81.59 (\#8) | 101.84 (\#11) |
> | LEDITS++ | SD1.5 | FP32 | 33.19 (\#6) | 45.18 (\#7) |
> | RF-Solver | FLUX.1-dev | BF16 | 256.23 (\#13) | 138.33 (\#12) |
> | RF-Inversion | FLUX.1-dev | BF16 | 117.24 (\#10) | 51.78 (\#8) |
> | FluxSpace* | FLUX.1-dev | BF16 | 145.50 (\#11) | 65.75 (\#9) |
> | FireFlow | FLUX.1-dev | BF16 | 29.85 (\#5) | 20.66 (\#2) |
> | FlowEdit | FLUX.1-dev | BF16 | 101.08 (\#9) | 43.95 (\#6) |
> | **FlowBypass** | SD3.5 Medium | FP16 | 29.43 (\#4) | 35.90 (\#5) |
> | **FlowBypass** | SD3.5 Large | FP16 | 64.11 (\#7) | 81.31 (\#10) |
> | **FlowBypass** | FLUX.1-dev | BF16 | 172.24 (\#12) | 192.23 (\#13) |
>
>
> We have included these new results and discussions in the revision.

---

> ### Author Response · Authors · 2025-11-21
> **Response 7EG6 (Part 4)**
>
> ## W5. Discussion on limitations
> Thanks for the reviewer’s suggestion. We agree that the discussion of limitations can be further strengthened.
>
> In the revision, we explicitly highlight the following limitations:
>
> **Moderate editing speed.**
> FlowBypass introduces additional overhead due to the calculation of bypass $b_t$.
> Although it is not the most computationally efficient method, the runtime of FlowBypass remains acceptable for single-image high-resolution editing.
>
> **Limited reliability on negation-based prompts (e.g., “without”).**
> Edits involving negation-based prompts sometimes fail. It is a common limitation that originates not from our framework but from the backbone generative models (FLUX / SD3.5), which are known to struggle with negative prompts. As a result, prompts such as *“a cat without a hat”* often still produce a cat wearing a hat. This issue is shared by many editing methods built upon these backbones, as the models themselves provide weak and unreliable outputs for negation. In practice, edits would become more stable and accurate when the negation is reformulated into direct phrasing without negation, such as transforming *“a cat with a hat”* into simply *“a cat”*.
>
> As for the performance in challenging scenarios, EditEvalv2 includes several challenging cases, such as extreme content modifications (object-removal sub-task) and style transfer edits (style-transfer sub-task), which are known to be difficult for training-free methods. Besides, we have added additional examples outside EditEvalv2 in the Appendix to illustrate the performance of FlowBypass under these challenging scenarios. Overall, FlowBypass demonstrates robust and reasonable performance even under these demanding conditions.
>
> We have included these observations in the revised version to more comprehensively present the applicability and limitations of FlowBypass.
>
>
>
> ## W6. Changing prompt for baselines
> We thank the reviewer for raising concerns about the fairness of prompt selection. We fully agree that prompt choice affects editing quality, as illustrated in Figure 6. Following the reviewer’s suggestion, we applied the same prompt choice ($C_y$ as positive prompt, $C_x$ as negative prompt during reconstruction, denoted as $yx$) used in FlowBypass to several baseline methods. The results in Table W6.1 show that while some baselines experience minor improvements, others suffer from failed edits and substantial performance degradation. This demonstrates that directly transferring FlowBypass’s prompt choice to other methods does not reliably improve performance and is not a feasible strategy for fair comparison.
>
> **Table W6.1** The impact of different prompt choice on baselines.
> | Method | Setting | LPIPS↓ | I.Sim.↑ | T.Sim.↑ |
> | ------- | ------- | ------- | ------- | ------- |
> | RF-Solver | Vanilla | 0.3880 | 87.32 | 25.30 |
> | RF-Solver | ee/yx | 0.3438 | 86.39 | 26.16 |
> | FireFlow | Vanilla | 0.3850 | 87.01 | 25.69 |
> | FireFlow | ee/yx | 0.3658 | 86.12 | 26.39 |
> | FlowEdit | Vanilla | 0.3921 | 87.90 | 25.28 |
> | FlowEdit | ee/yx | 0.2087 | 97.71 | 22.62 |
>
>
> Our prompt selection in FlowBypass is not an empirical hyperparameter tuning but is dictated by theoretical derivation. Specifically, in the analytical solution of a first-order linear differential equation, the exponential term acts as the accumulated effect of nonlinear terms along the trajectory. In FlowBypass, this exponential term compensates for the semantic discrepancies introduced by the Taylor expansion. Consequently, using the "yx" prompt configuration provides a balanced compensation for this integrating term and amplifies the semantic shift from origin semantics to edit semantics.
> Alternative settings (*e.g.*, "ye" and "yy") would create an imbalanced compensation, causing the bypass computation to overly favor either the target or the origin image, thereby harming fidelity or alignment.
>
> Other methods, such as FlowEdit, do not explicitly decompose the compensation term and therefore do not require this balancing. Directly applying FlowBypass’s prompt strategy to these methods is not only ineffective but may also disrupt their original mechanism. Our experiments in Table W6.1 confirm this: the same prompt strategy does not consistently improve baseline performance and can even lead to significant editing failure.
>
> Therefore, we believe that the current comparison is fair and will not introduce bias due to prompt settings.
>
> We hope this clarifies the fairness concern and highlights the principled motivation behind FlowBypass’s prompt selection. We have clarified this in the revision.

---

> ### Author Response · Authors · 2025-11-21
> **Response 7EG6 (Part 5)**
>
> ## Q1. Ablation study on prompt choice
> We thank the reviewer for the suggestion. Figure 6 and Table 4 presents ablation studies for different prompt combinations. The setting suggested by the reviewer (leaving the negative prompt empty) corresponds to the ee/ye combination. This configuration completely omits the origin prompt $C_x$, which is why it was not included in the initial submission. We have now added this experiment in the revision.
>
> **Table Q1.1** Impact of prompt choice. [F] indicates the result from FLUX.1-dev, [S] indicates the result from SD3.5 Large.
> | Setting | [F] LPIPS↓ | [F] I.Sim.↑ | [F] T.Sim.↑ | [S] LPIPS↓ | [S] I.Sim.↑ | [S] T.Sim.↑ |
> | ------- | ------- | ------- | ------- | ------- | ------- | ------- |
> | ee/ye | 0.3586 | 89.65 | 25.80 | 0.4716 | 85.55 | 27.14 |
> | ee/yy | 0.2685 | 93.16 | 24.38 | 0.3759 | 90.00 | 25.80 |
> | **ee/yx** | 0.3425 | 88.06 | 25.65 | 0.4507 | 84.73 | 27.09 |
>
>
> The results show that omitting the original prompt $C_x$ as the negative prompt during reconstruction degrades editing performance. This occurs because $C_{rec}^n = C_x$ provides the model with essential semantic constraints from the origin prompt. Without it, the semantic path from inversion to reconstruction becomes unbalanced, causing the model to overemphasize the edit prompt, which leads to over-editing and low-fidelity. On the other hand, the ee/yy (i.e., disabling CFG in reconstruction) leads to editing failure, as the absence of CFG significantly degrades reconstruction quality.
>
> Therefore, using $C_{rec}^p = C_y$ and $C_{rec}^n = C_x$ is crucial for maintaining reconstruction stability and ensuring editing fidelity.
>
> We hope these supplementary experiments can help the motivation and effectiveness of our prompt choice.
>
> ## Q2. Ablation on $\zeta$ and exponential approximation
> We thank the reviewer for the suggestion. We have added an ablation study on the hyperparameter $\zeta$ in the revision (see the Table Q2.1, **Table 9 in the revision**). The results indicate that FlowBypass is reasonably robust to $\zeta$; varying it within a practical range does not meaningfully influence editing quality.
>
>
> **Table Q2.1** Impact of $\zeta$. [F] indicates the result from FLUX.1-dev, [S] indicates the result from SD3.5 Large.
> | $\zeta$ | [F] LPIPS↓ | [F] I.Sim.↑ | [F] T.Sim.↑ | [S] LPIPS↓ | [S] I.Sim.↑ | [S] T.Sim.↑ |
> | ------- | ------- | ------- | ------- | ------- | ------- | ------- |
> | 0.001 | 0.3290 | 88.56 | 25.66 | 0.4513 | 84.77 | 27.10 |
> | 0.005 | 0.3446 | 87.91 | 25.77 | 0.4488 | 85.09 | 27.14 |
> | **0.01** | 0.3425 | 88.06 | 25.65 | 0.4507 | 84.73 | 27.09 |
> | 0.05 | 0.3509 | 88.03 | 25.84 | 0.4606 | 84.65 | 27.21 |
> | 0.1 | 0.3434 | 88.25 | 25.82 | 0.4562 | 84.81 | 27.32 |
>
> Regarding the suggestion of “replacing the exponential with its first-order Taylor approximation in the positive domain”, this experiment has been reported in Table 3 and Figure 8, which has been reported as **w/o Approx. exp**. The results show that removing the approximation that replace the exponential with its first-order Taylor approximation in the positive domain leads to noticeably stronger over-editing and instability. In contrast, the first-order approximation successfully suppresses the amplification effect of the exponential term while preserving performance, making it a more stable choice.
>
> These findings demonstrate that both design components in Equation 13 have been empirically validated through ablations, rather than being arbitrarily chosen.
>
> ## Q3. Visualization of bypass $b_t$ under different $t$
> Thank you for the helpful suggestion. We would like to clarify that FlowBypass performs **only one** bypass computation and transition during an actual editing process. For the purpose of requested visualization, we additionally compute bypass variables under different choices of $t_B$ within the same editing run, solely to illustrate how the bypass behaves when applied at different bypass timesteps, as illustrated in **Figure 16 of the revision**.
>
> The newly added visualizations show that larger values of $t_B$ (i.e., earlier denoising stages) tend to influence global layout and structure, while smaller values of $t_B$ (i.e., later stages) exhibit the modification of local details and texture refinement. This pattern aligns well with observations reported in prior works on DDIM and RF-based sampling, and provides an intuitive view of how the bypass mechanism modulates semantic corrections across the denoising trajectory.
>
> Regarding the noisy intermediate states $y_{t_B}$, their early-stage representations are dominated by noise and are nearly uninformative. As such, visualizing them would not meaningfully contribute to explaining the editing mechanism.
>
> We hope that visualizing the bypass mechanism across different values of $t_B$ will help reviewers better understand its behavior and impact.

---

> ### Author Response · Authors · 2025-11-21
> **Response 7EG6 (Part 6)**
>
> ## Q4. Computational cost breakdown and runtime comparison
> Under our default configuration, a full editing pass requires **190 model forward evaluations**. This corresponds to an overhead of approximately **21.1%** compared with the standard inversion–reconstruction pipeline theoretically, primarily due to the additional inference steps needed to compute $b_{t_B}$.
> We break down the computational cost into various stages, as shown in Table Q4.1 (**Table 12 in the revision**). The discrepancy between the totals in Table Q4.1 and Table W4.2 arises because the text encoding time is excluded in Table Q4.1.
>
> **Table Q4.1** Computational cost breakdown (second). The % in parentheses indicates the percentage.
> | Stage | SD35M | SD35L | FLUX.1 |
> | ------- | ------- | ------- | ------- |
> | VAE encode | 0.23 (1.08%) | 0.16 (0.31%) | 0.22 (0.13%) |
> | Inversion | 9.71 (46.25%) | 24.38 (46.97%) | 78.50 (46.52%) |
> | **Inversion for Bypass** | 4.32 (20.56%) | 10.84 (20.87%) | 34.89 (20.67%) |
> | **Bypass** | 0.00836 (0.0398%) | 0.00335 (0.006461%) | 0.00359 (0.002129%) |
> | Recon | 6.42 (30.56%) | 16.27 (31.34%) | 54.90 (32.53%) |
> | VAE decode | 0.32 (1.51%) | 0.26 (0.50%) | 0.24 (0.14%) |
>
> Notably, these additional computations are causally independent, enabling parallelization across multiple timesteps via batching. Consequently, the actual wall-clock time can be lower than a purely serial estimation would imply.
>
> In the revision, we also include a runtime comparison against several baseline methods (see Table W4.2 in Weakness 4). The results show that although FlowBypass introduces some overhead, the overall computational cost remains within a reasonable range, while delivering superior editing quality.

---

### Author Response · Authors · 2025-11-21
**General Response**

## General Response
We sincerely thank all reviewers for their thoughtful, detailed, and constructive feedback!

We are pleased that all reviewers (7EG6, K56Z, z99e, bETU) recognize the significance and generalizability of our training-free trajectory bypass framework for inversion-based image editing:

* *“The training-free image editing framework … seems shows strong generalizability across Rectified Flow models.”* — **7EG6**
* *“The method effectively addresses the fidelity-alignment trade-off without relying on model-specific feature manipulations, ensuring broader applicability.”* — **K56Z**
* *“The proposed method is not tied to specific backbone architectures, which can be applied on various base models such as FLUX and SD3.”* — **z99e**
* *“By abstaining from backbone-specific Feature Manipulation (FM), the method successfully demonstrates robust performance across diverse Rectified Flow architectures (SD3.5M/L and FLUX.1-dev). This general applicability makes FlowBypass a potentially influential technique in the broader field of diffusion model editing.”* — **bETU**

We are also glad that the reviewers (7EG6, K56Z, bETU) point out our theoretical formulation rigorous and principled:

* *“The core idea appears theoretically sound, ...”* — **7EG6**
* *“The core idea of constructing a bypass between inversion and reconstruction trajectories is well-motivated. The theoretical derivation of the bypass term from first principles is rigorous and elegant.”* — **K56Z**
* *“Strong and Novel Theoretical Foundation: ... This principled approach provides a solid, non-heuristic framework for error mitigation in inversion-based editing, which is a significant theoretical advance over prior empirical or feature-manipulation-based methods.”* — **bETU**

We are further pleased that all reviewers (7EG6, K56Z, z99e, bETU) highlight the strong performance and thorough analysis in our work:

* *“The ..., and the experimental results demonstrate promising performance.”* — **7EG6**
* *“The method effectively addresses the fidelity-alignment trade-off without relying on model-specific feature manipulations, ensuring broader applicability.”* — **K56Z**
* *“Quantitative and qualitative results in paper demonstrate that the outstanding performance of the proposed method.”* — **z99e**
* *“Competitive State-of-the-Art Performance: ...  It achieves superior perceptual fidelity (lowest LPIPS on FLUX.1-dev) and strong text alignment (best T.Sim. on SD3.5L), confirming its effectiveness in achieving the intended balance.”* — **bETU**
* *“Thorough Ablation Studies: The authors provide comprehensive ablation studies on critical design choices, ... These experiments solidify the robustness and rationale behind the proposed framework's final configuration.”* — **bETU**


## Updates and New Experiments
We summarize the additional analyses and experiments inspired by the reviewers’ comments. These updates will be incorporated into the revised PDF after further discussion with reviewers.

* **More compared baselines:**
We introduce four additional baselines for a broader and more comprehensive comparison, including **FlowEdit**, **FluxSpace**, **Prompt-to-Prompt**, and **FireFlow**. The quantitative results have been updated in **Table 1** of the revision, and the qualitative comparisons have been updated in **Figure 2** and **Figure 11**. Besides, we conduct a user study to evaluate how well FlowBypass aligns with human preference, as presented in **Sec. A.3.2**.

* **Runtime comparison with SOTA methods and runtime breakdown:**
We provide a detailed runtime comparison in **Table 7** of the revision, along with a breakdown and analysis of the time cost at different stages in **Table 8**.

* **Additional ablation studies and visualizations:**
We report further ablation studies, including the impact of the effect of different prompt choices (**Sec. 4.4.3**), the hyperparameter $\zeta$ (**Sec. A.3.6**), and edit performance when editing starts from various intermediate timesteps without the bypass (**Sec. A.3.8**). In addition, we provide more visualizations, including extra editing results beyond the EditEvalv2 dataset in **Figure 13** and visualizations of the bypass under different $t_B$ in **Figure 16**.

* **More analysis and expanded discussion of limitations:**
We refine the analysis and interpretation of our results, and we add a dedicated **Sec. 5** to discuss the limitations of FlowBypass.

We sincerely thank all reviewers again for their insightful suggestions and look forward to further discussion any time.

---

### Author Response · Authors · 2025-12-03
**Summary for Area Chair**

**Dear Area Chair,**

In light of the recent AC re-assignment, we are writing to provide a concise summary of the rebuttal discussion and the corresponding updates to our paper. We sincerely appreciate the insightful comments from the reviewers and the efforts of the area chairs throughout the review process, and we have made every effort to thoughtfully address all concerns they have raised.

**General Reception**

We were encouraged that reviewers recognized our work’s key strengths, describing it as “well-motivated” (**K56Z**) with a “strong and novel theoretical foundation” (**bETU**), supported by “rigorous and elegant” derivation (**K56Z**), demonstrating “the outstanding performance” (**z99e**), and “seems shows strong generalizability across Rectified Flow models” (**7EG6**), ultimately highlighting FlowBypass as “a potentially influential technique in the broader field of diffusion model editing” (**bETU**).


**New Experiments & Baselines**

In response to requests for additional experiments and baselines, we conducted extensive new experiments:

* **More compared baselines**: We added four additional baselines for both quantitative and qualitative comparisons. *FlowBypass consistently outperforms all newly introduced baselines, achieving strong alignment without noticeably compromising fidelity*. (Table 1, Figure 2, Figure 11 and Sec. A.3.2 in the revision)

* **Runtime comparison with SOTA methods and runtime breakdown**: We included a comprehensive runtime comparison against SOTA methods, along with a detailed breakdown and analysis of the time cost at each stage. *FlowBypass attains competitive efficiency, with bypass computation introducing only a modest overhead.* (Table 7 and Table 8 in the revision)

* **Additional ablation studies and visualizations**: We expanded our ablation studies to cover prompt choices, *demonstrating the superiority of designed prompt choice*; the hyperparameter $\zeta$, showing *the robustness of $\zeta$ within FlowBypass*; and editing without bypass at various intermediate timesteps, revealing *the necessity of the bypass*. We also provided additional visualizations to qualitatively showcase *the strong editing performance*. (Sec. 4.4.3, Sec. A.3.6, Sec. A.3.8, Figure 13 and Figure 16 in the revision)

**Key Clarifications**

We also addressed specific concerns raised during the review process, particularly those regarding our notation, the distinctions against prior approaches, the validity of approximation, and the limitations:

* **The notation**: We refined and clarified the notation throughout the manuscript to improve clarity and readability. (Response to Weakness 1 from 7EG6)

* **Distinctions against prior approaches**: We elaborated on the key differences between FlowBypass and previous methods, highlighting both the conceptual distinctions and the advantages of our approach. (Response of Weakness 2 from 7EG6)

* **Validity of approximation**: We clarified the behavior of the bypass and the conditions under which the first-order Taylor approximation may lose accuracy, explained its practical reliability in typical settings, and supported this with an ablation on the hyperparameter $\zeta$. (Response to Weakness 3 from 7EG6)

* **Limitation**: We discussed the limitation of FlowBypass, and provided possible solution for the limitation. (Responses of Weakness 5 from 7EG6, Weakness 2 from K56Z)

Thank you for your time and supervision of the review process.

Sincerely,

The Authors of Submission 3283

---

### Meta-Review · Area_Chair_jEsH · 2025-12-31

**Summary:**

In the initial reviews, 3 reviewers recommend rejection and one recommends acceptance. There was also a short initial discussion (truncated due to the OpenReview security incident).

Reviewer 7EG6's recommends rejection. They are unsure what the conceptual differences between the proposed approach and FlowEdit are and feel that the derivation in Eq 5-7 should more clearly discuss this prior work. They also raise concerns with how prompts were used in the proposed method (which requires a combination of 4 prompts) versus the baselines. The authors conducted an ablation study to study the impact of prompt choice. The experiments suggest that transferring these prompts to other methods does not improve them. Reviewer K56Z recommends acceptance, finding the core idea to be well motivated. They raise concerns with the heavy reliance on EditEvalv2 and automated benchmarks. The authors conducted a user study to address this, which finds that users prefer their image edits over baselines on several different models. Reviewer z99e recommends rejection and asks for more comparisons to baselines, but finds that the method is well motivated and appreciates that it can be applied to several different generators (they also have a few questions about possible changes to the method). The authors address this by providing more comparisons and by doing a runtime evaluation. Reviewer bETU recommends rejection. They appreciate the theoretical motivation, the generality of the approach (it doesn't rely on a specific network architecture so it works on several backbones), and the thoroughness of the evaluation (in contrast to other reviewers). However, they raise concerns about the finite difference approximation of the gradient (in the rebuttal, the authors point out that the overhead is about 20%) and about the accuracy of the approximations used in the derivation (including a Taylor expansion). The authors address this through a new ablation experiment.

On balance, the AC feels that the clarity issues and the evaluation concerns raised by reviewer 7EG6 outweigh the benefits of the approach and feels that the paper requires revision before it can be accepted.

**Reviewer Concerns:**

The authors submitted an extensive rebuttal that addresses many of the reviewer points through new experiments.

For Reviewer 7EG6, I think that the rebuttal does a reasonable job of explaining the high level differences between their approach and FlowEdit. The reviewer (as part of the discussion) raised concerns that the paper does not credit FlowEdit sufficiently. They still have concerns with fairness of the comparison to prior methods, related to the prompt selection.

Reviewer K56Z's concerns about the user study would likely be partly addressed by the rebuttal's new experiments, though there may be remaining methodological issues (due to the difficulty of conducting such as study). They may also have remaining concerns about the extensiveness of the evaluation (given their worries about the use of 150 test images).

Reviewer z99e's concerns about additional comparisons and questions about running time may have been addressed by the rebuttal.

Reviewer bETU's concerns about computational overhead may be addressed, since the authors report a modest (20%) theoretical overhead; it is less clear whether the sensitivity to hyperparameters and approximation accuracy concerns would be satisfactorily addressed, though the authors attempt to address them through experiments.

**Reviewer Scores:**

- Reviewer 7EG6: would likely have maintained their rating (4).
- Reviewer K56Z: would likely have maintained their rating (6).
- Reviewer z99e: may have increased their rating (from 4) and recommended acceptance.
- Reviewer bETU: it is not clear how they would have changed their score. I think it's equally likely they would have increased or maintained their score.

---

### Decision · Program_Chairs · 2026-01-26

Reject